# Enabling giant thermopower by heterostructure engineering of hydrated vanadium pentoxide for zinc ion thermal charging cells

Zhiwei Li [1], Yinghong Xu[1], Langyuan Wu[1], Jiaxin Cui[1], Hui Dou[1] & Xiaogang Zhang [1] ✉

Flexible power supply devices provide possibilities for wearable electronics in the Internet of Things. However, unsatisfying capacity or lifetime of typical batteries or capacitors seriously limit their practical applications. Different from conventional heat-to-electricity generators, zinc ion thermal charging cells has been a competitive candidate for the self-power supply solution, but the lack of promising cathode materials has restricted the achievement of promising performances. Herein, we propose an attractive cathode material by rational heterostructure engineering of hydrated vanadium pentoxide. Owing to the integration of thermodiffusion and thermoextraction effects, the thermopower is significantly improved from $7.8 \pm 2.6$ mV K$^{-1}$ to $23.4 \pm 1.5$ mV K$^{-1}$. Moreover, an impressive normalized power density of 1.9 mW m$^{-2}$ K$^{-2}$ is achieved in the quasi-solid-state cells. In addition, a wearable power supply constructed by three units can drive the commercial health monitoring system by harvesting body heat. This work demonstrates the effectiveness of electrodes design for wearable thermoelectric applications.

Portable devices are widely used in wearable applications for communication, health monitoring, and other areas. However, the unsatisfying capacity and lifetime of power systems like batteries and supercapacitors seriously hinder the development of wearable electronics[1,2]. Under this consideration, power supply systems with characteristics of low-cost, high performance, sustainability, and good durability are extremely necessary for the construction of Internet of Things (IoTs)[3–6]. Thermoelectric devices (TEs), as one of the key techniques, can directly convert low-grade heat into electricity under a low temperature difference of several Kelvins based on the Seebeck effect, which provides in principle the practicability of TEs in self-power supply for electronics using human body heat[7–10]. Very recently, Qiu and his co-workers discovered a series of p-type ductile TE materials, AgCu(Se, S, Te) pseudoternary solid solutions for the fabrication

of flexible TEs[11]. When coupling with the n-type ductile material (Ag$_{20}$S$_7$Te$_3$), they developed a π-type flexible TE devices with a thickness of 0.3 mm and high normalized power density of 30 μW cm$^{-2}$ K$^{-2}$. It should point out that this device-integrated 31-pair flexible TE units adhered to human's wrist can output a voltage of 0.2 mV and power of 70 nW. Although some applied heat-dissipation routes like copper pipes and graphite sheets would further improve the output performances to power some microelectronics, the relatively complicated integration of working units still is the key issue that needs to be addressed.

Alternatively, ionic thermoelectric devices (i-TEs) are proposed for low-grade heat harvesting by combining the diffusion of ions and redox reactions of species under the existence of temperature gradient[12,13]. Owing to the rational integration of thermodiffusion and

---

[1]Jiangsu Key Laboratory of Electrochemical Energy Storage Technologies, College of Material Science and Technology, Nanjing University of Aeronautics and Astronautics, Nanjing 211106, China. ✉e-mail: azhangxg@nuaa.edu.cn

thermogalvanic effects, i-TEs have been widely concerned by researchers in practical applications[14]. As a proof-of-concept, Liu et al. developed a new-type ionic thermoelectric material for the harvesting of human body heat[15]. Typically, potassium chloride (KCl) and ferro/ferricyanide [$Fe(CN)_6^{4-}$/$Fe(CN)_6^{3-}$] were optimized in a gelatin matrix to simultaneously achieve the synergistic changes of thermodiffusion entropy and thermogalvanic entropy. As a result, a giant thermopower of $17.0\,mV\,K^{-1}$ can be obtained using copper electrode under a temperature difference of ~10 K. Most importantly, a high voltage of 2 V and a peak power of 5 mW can be generated from the harvesting of body heat by integrating 25 unipolar units. Huang et al. designed a nonaqueous ionogel with strong ion-ion interactions introduced by selective ion doping to adjust the ionic thermopower[16]. Detailly, ionogels based on an ionic liquid (EMIMTFSI/PVDF-HFP) shows high negative thermopower of $-15\,M\,K^{-1}$ after adding $0.5\,M\,LiBF_4$. Moreover, a peak value of positive thermopower ($17\,mV\,K^{-1}$) can be achieved with the introduction of EMIMCl. According to these inspired results, the prototype device with 12 pairs of n-/p-type units, which is adhered on human arm, can produce a voltage of 0.33 V. Apart from the development of advanced ionogels, the introduction of chaotropic/host agent is an efficient strategy to enhance the thermoelectrochemical performances by increasing the concentration difference[17]. For example, Zhou and co-workers reported a thermosensitive guanidinium ($Gdm^+$) induced $Fe(CN)_6^{4-}$ crystallization to construct high-performance thermocells[18]. With the precipitation of crystals, both the voltage and thermopower of the system can be enhanced. However, it is difficult to adopt the precipitation and redissolution of thermosensitive crystallization into wearable applications that would meet the demands of multidimensional motion. Till now, the development of high-performance thermoelectric devices has been the top priority for self-powered electronics through human body heat harvesting.

Besides the optimization of functional electrolytes, electrode engineering is another efficient approach to improve thermoelectrochemical performances[19]. Based on the electrical double-layer capacitive effect or near-surface redox behavior, some electrode materials including porous carbon, carbon nanotube, and conductive polymer are widely studied in previous reports[20–26]. To further boost the energy conversion efficiency and energy storage density during the thermoelectrical process, we have proposed a zinc ion thermal charging cell (ZTCC) using Zn metal anode and vanadium dioxide-porous carbon (VO2-PC) cathode[27]. As-fabricated ZTCC can convert low-grade heat into electricity with a thermopower of ~$12.5\,mV\,K^{-1}$ by the combination of thermodiffusion and thermoextraction. Notably, the whole thermoelectrochemical process is dominated by the thermodiffusion effect, which may be caused by the embedding feature of $VO_2$ in PC matrix. It should be mentioned that the average ion diffusion coefficient of VO2-PC is in a relatively low level ($3.16 \times 10^{-11}\,cm^2\,S^{-1}$). To realize a fast thermoelectrochemical response of ZTCCs, the kinetics matching of themodiffusion and thermoextraction, as well as the ion mobility and conductivity of electrode materials, should be well considered. Importantly, vanadium oxides (especially for vanadium pentoxides) are regarded as attractive candidates for energy storage and conversion owing to their relatively large inner spacing, tuned nanostructures, and multielectron reactions[28,29]. Nonetheless, the dissolution of vanadium in neutral or acidic electrolytes during electrochemical processes would result in the failure of electrodes and contamination of electrolytes. Meanwhile, the low electronic conductivity and electrostatic repulsion of layered vanadium oxides always lead to unsatisfying rate/power performances.

Herein, we propose and demonstrate a ZTCC with boosted performances using graphite modified zinc anode (Zn-G) and vanadium pentaoxide@reduced graphene oxide ($V_2O_5$@rGO) cathode. In detail, the properties of $V_2O_5$ are reasonably adjusted by the addition of rGO. Benefitting from the thermodiffusion contribution of rGO and

thermoextraction behavior of $V_2O_5$, high-performance ZTCC in term of thermopower and power density can be achieved. Worthily, the rGO nanosheets can not only enhance the kinetics of $V_2O_5$ host, but also suppress the dissolution of vanadium species. Such attractive integration between $V_2O_5$ and rGO endows as-prepared $V_2O_5$@rGO composites with high thermal and electrochemical stability. In addition, the $V_2O_5$@rGO-1.5 based ZTCC shows great promise in the application of wearable health monitoring systems. Compared with other strategies for designing flexible thermoelectric systems, this work provides a facile way to construct a boosted ionic thermoelectrochemical device by the efficient structure regulation of electrode materials.

## Results

### Boosted ZTCCs enabled by $V_2O_5$@rGO-x

ZTCC constructed by insertion-type cathode and zinc metal anode attracts great attention due to the promising thermoelectrochemical performances during the conversion of low-grade heat and energy storage. As schematically illustrated in Fig. 1a, the thermopower of ZTCC is mainly determined by the thermodiffusion of electrolyte ions and thermogalvanic processes of electrodes. In detail, the pre-inserted cations in the cathode can be gradually extracted with heat input and transferred to the anode side under the effect of the thermal field. With the plating of diffused $Zn^{2+}$ in Zn anode, the heat energy can be converted into electricity and stored simultaneously. Fig. 1b displays the possible voltage curve. To achieve a boosted thermoelectrochemical response, the development of electrode materials with fast kinetics and good durability becomes significant. The combination of vanadium oxides with nanocarbons shows great promise to overcome such challenges. As a demo, we proposed the $V_2O_5$·$1.6H_2O$ with large interlayer spacing as a host material for charge storage and rGO as a functional matrix to enhance the kinetics and stability of composites. The density functional theory (DFT) calculations were performed on the model structure to understand the relationship between the electronic structure of $V_2O_5$@rGO and the superior kinetics behavior. As shown in Fig. 1c, the differential charge density of $V_2O_5$@rGO displays obvious charge accumulation around the V and O atoms and dispersion of surrounding electron states around the Zn atom. This charge redistribution reveals the optimized electronegativity of $V_2O_5$·$1.6H_2O$, further boosting the adsorption ability to $Zn^{2+}$. Consequently, the $V_2O_5$@rGO exhibits a low negative adsorption energy of $Zn^{2+}$ (−1.47 eV). This suggests that the ultra-large interlayer spacing of $V_2O_5$·$1.6H_2O$ is beneficial to the adsorption and diffusion of $Zn^{2+}$. Simultaneously, the crystal $H_2O$ molecules in $V_2O_5$ layers can act as a "lubricant" to improve host electrochemical kinetics. Subsequently, the rGO layer on $V_2O_5$·$1.6H_2O$ surface retains structural integrity and accelerates the charge transfer. In addition, the diffusion pathway of $Zn^{2+}$ in the $V_2O_5$@rGO sample is shown in Fig. 1d. It is worth mentioning that the energy barrier for ions diffusion in $V_2O_5$@rGO is at a low level of 0.27 eV (Supplementary Fig. 1), confirming the easy diffusion of $Zn^{2+}$ along the $V_2O_5$@rGO heterointerface. Such high $Zn^{2+}$ diffusion coefficient and low diffusion barrier endow $V_2O_5$@rGO composites with high-power/rate performances in energy conversion and storage applications.

According to the guidelines, we propose an electrostatic interaction-induced self-assembly strategy to realize the structure regulation on the $V_2O_5$@rGO-x nanocomposites (x: concentration of GO solution, mg $mL^{-1}$). The field-emission scanning electron microscopy (FESEM) and transmission electron microscopy (TEM) images in Supplementary Fig. 2 and Fig. 1e clearly indicate that the $V_2O_5$@rGO-1.5 shows reasonably intertwined microstructure engineered by multilayer rGO and $V_2O_5$ nanobelts with several micrometers in length. It is worth mentioning that the 1D $V_2O_5$ nanobelts with large interlayer spacing can provide abundant electrochemical sites for ions intercalation/deintercalation and the 2D rGO nanosheets wrapped around the $V_2O_5$ nanobelts can act as matrix for ions adsorption/desorption as

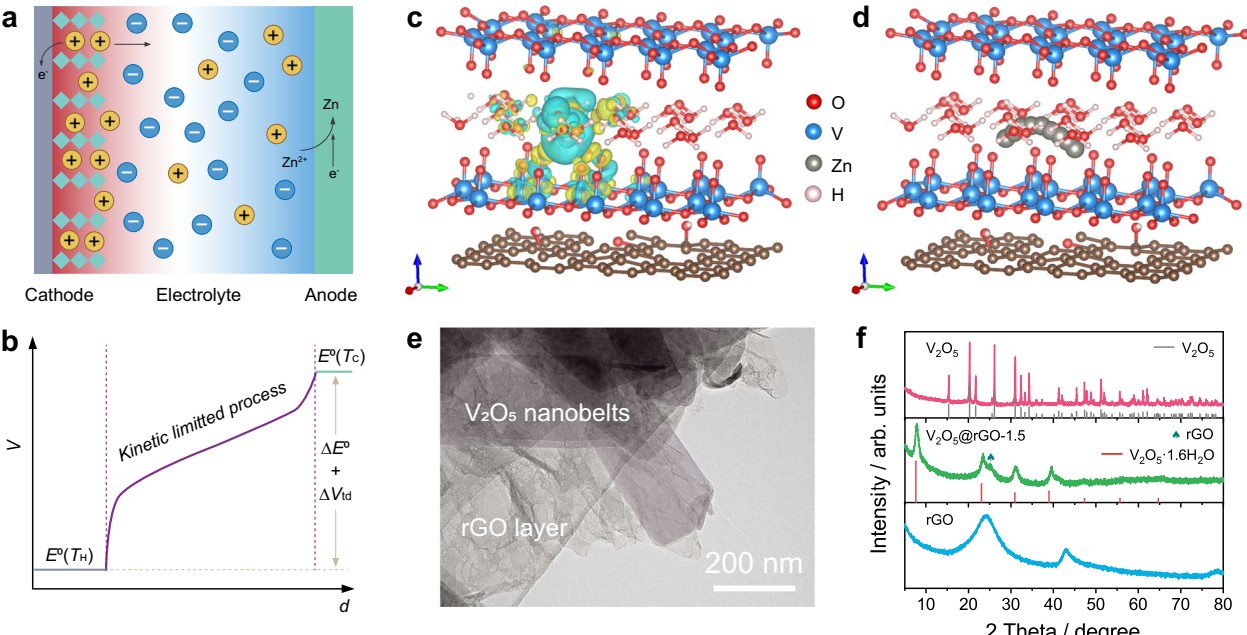

**Fig. 1 | Construction of boosted ZTCC and characterization of V₂O₅@rGO sample. a** Schematic illustration of V₂O₅@rGO based ZTCC, and **b** corresponding voltage distribution profile. **c** Differential charge density, and **d** the optimized diffusion pathway of Zn²⁺ in V₂O₅@rGO composite. **e** TEM image of V₂O₅@rGO-1.5. **f** XRD pattern of V₂O₅, V₂O₅@rGO-1.5, and rGO.

well as a pathway for charge transfer. However, the pure V₂O₅ obtained without adding GO solution exhibits random nanoparticle morphology (Supplementary Fig. 3a), which is significantly different from the nanobelt-type V₂O₅ in V₂O₅@rGO nanocomposites. Moreover, the pristine GO after hydrothermal process displays a well-developed 3D continuous framework formed by corrugated and stacked rGO nanosheets (Supplementary Fig. 3b). When considering the fully different microstructures of V₂O₅@rGO, V₂O₅, and rGO samples, it can be concluded that the introduction of GO into the preparation of V₂O₅ is a rational approach to regulate the structure of nanohybrids. In fact, the relatively acidic atmosphere (pH: ~3.0) caused by GO solution could promote the generation of positively vanadium oxide, which would adsorb on the negatively charged surface of GO and gradually evolve into nanobelts between rGO layers[30]. The morphologies of V₂O₅@rGO composites highly depend on the content of rGO. As displayed in Supplementary Fig. 3c, the V₂O₅@rGO-1.0 holds the characteristics of V₂O₅ nanobelts, in which rGO substrate is fully covered. While for the V₂O₅@rGO-2.0, only a few V₂O₅ nanobelts are exposed from the 3D graphene network (Supplementary Fig. 3d). The corresponding high-resolution transmission electron microscopy (HRTEM) image in Supplementary Fig. 4 confirms the successful synthesis of layered V₂O₅ in V₂O₅@rGO-1.5 sample. Consistently, the ultra large interlayer spacing of around 1.1 nm can be indexed to the (001) plane of orthorhombic V₂O₅·1.6H₂O[31]. Selected area electron diffraction (SAED) pattern reveals the polycrystalline feature of formed V₂O₅·1.6H₂O nanobelts and amorphous rGO layers (Supplementary Fig. 5), further confirming the formation of heterostructure by V₂O₅·1.6H₂O and rGO. Meanwhile, the high-angle annular dark-field scanning transmission electron microscopy (HAADF-STEM) and corresponding elemental mapping images detect the distribution of V, C, and O elements in V₂O₅@rGO-1.5, demonstrating the rational integration of rGO and V₂O₅ (Supplementary Fig. 6).

The phase and structure of samples are confirmed by the X-ray diffraction (XRD) patterns. As plotted in Fig. 1f, all the obvious peaks of V₂O₅@rGO-1.5 correspond to the characteristic peaks of orthorhombic V₂O₅·1.6H₂O (PDF#40−1296). The additional peak around 25.3° in the spectrum of V₂O₅@rGO-1.5 can be attributed to the (002) plane of amorphous rGO. Interestingly, the pure V₂O₅ obtained without addition of GO is in terms of orthorhombic V₂O₅ (PDF#89-0612)[32]. Notably, the pattern of V₂O₅@rGO-1.0 is similar with that of V₂O₅@rGO-1.5 (Supplementary Fig. 7). The interlayer spacing of pure V₂O₅ is significantly enlarged from ~0.44 nm to ~1.1 nm of V₂O₅·1.6H₂O in both V₂O₅@rGO-1.5 and V₂O₅@rGO-1.0, implying that acidic GO exhibits important role in regulating the microstructure of vanadium oxides. Moreover, as detected by the XRD pattern of V₂O₅@rGO-2.0, the V₂O₅·1.6H₂O crystals can further evolve into low-valence vanadium species like V₆O₁₃ and V₃O₇·1.6H₂O with the increasing of GO concentration due to partial reduction of high-valence vanadium species by formed rGO during hydrothermal treatment. The comparison of Raman spectra in Supplementary Fig. 8 shows typical characteristics of rGO (1330.7 and 1598.2 cm⁻¹) and VO$_x$ (142.5, 194.4, 662.2, 404.8, 698.2 and 992.7 cm⁻¹)[33,34]. The co-existence of above Raman peaks in V₂O₅@rGO-x hybrid demonstrates the rational combination of rGO and VO$_x$, further confirming the composition of the heterostructure. Moreover, the thermogravimetric analysis (TGA) indicates that the V₂O₅ in V₂O₅@rGO-1.5 contains ~1.62 crystal water molecules per unit, and the content of rGO is about 31.7% (Supplementary Fig. 9). In addition, the specific surface area (SSA) of sample is measured by N₂ adsorption-desorption isotherms (Supplementary Fig. 10a). Benefitting to the abundant pores and 3D-interconnected structure, the rGO shows the largest adsorbed quantity among as-prepared materials. Consequently, a high SSA value of 284.1 m² g⁻¹ can be achieved (Supplementary Fig. 10b). However, the pristine V₂O₅ only shows 5.2 m² g⁻¹. After introducing rGO, the SSA value can be reasonably improved to 13.3 m² g⁻¹ for V₂O₅@rGO-1.5. Such high SSA together with mesoporous feature of V₂O₅@rGO-1.5 can effectively boost the ions mobility. The X-ray photoelectron spectroscopy (XPS) of V₂O₅, rGO and V₂O₅@rGO-x composites was carried out to reveal the integration of carbon matrix and vanadium oxides (Supplementary Fig. 11a). Notably, the broad peak of O 1s for V₂O₅ at 530.3 eV can be attributed to the VO$_x$ (Supplementary Fig. 11b)[35]. Deconvolution of O 1s spectra for V₂O₅@rGO-x samples shows two additional peaks at 531.4 and 533.4 eV, which corresponds to the C = O groups in rGO and the water in the sample, respectively[35,36]. As shown in Supplementary Fig. 11c, the peaks at 517.3

(V $2p_{3/2}$) and 524.6 eV (V $2p_{1/2}$) for $V_2O_5$ belong to $V^{5+}$[37]. After combining with rGO, a part of $V^{4+}$ can be found in V $2p$ of $V_2O_5$@rGO-x composites. Detailly, the ratio of $V^{4+}$ to $V^{5+}$ gradually increases from 0.28 ($V_2O_5$@rGO-1.0) to 0.47 ($V_2O_5$@rGO-2.0), which could be caused by the reduction effect of rGO to $V^{5+}$ during the hydrothermal procedure. Moreover, the contact angle (CA) of pristine $V_2O_5$ is about 26.1° (Supplementary Fig. 12a), suggesting its superhydrophilic feature. Even after standing 20 s, the CA value still can be maintained at 23.2° (Supplementary Fig. 12b). Worthily, the CA value of $V_2O_5$@rGO-x composite gradually increases with the increase of rGO amount, which may be caused by the stack of graphene layers. According to the change of CA values during electrolyte immersion, the $V_2O_5$@rGO-1.0 delivers higher electrolyte permeability than that of $V_2O_5$@rGO-1.5 and $V_2O_5$@rGO-2.0. These differences in CA values highly affect to the dissolution of vanadium species from the electrode into the electrolyte. As shown in Supplementary Fig. 13a, the electrolyte solution with pristine $V_2O_5$ shows a noticeable color change from transparent to yellow, suggesting the serious dissolution of V species in the electrolyte. As expected, the dissolution of V species of various $V_2O_5$@rGO-x electrodes can be significantly suppressed by the rGO coating layers. Furthermore, the detailed concentration of V species in electrolytes after soaking for 5 days and 10 days is summarized in Supplementary Fig. 13b. After soaking for 5 days, the concentration of V in electrolyte for $V_2O_5$ is 47.5 mg L$^{-1}$, which is several times higher than those for $V_2O_5$@rGO-x electrodes (8.9 ~ 17.3 mg L$^{-1}$). Even after soaking for 10 days, the average V element dissolution rate of $V_2O_5$ (7.1 mg L$^{-1}$ per day) is much higher than those of $V_2O_5$@rGO-x electrodes

(1.1 ~ 2.0 mg L$^{-1}$ per day), confirming the important role of rGO to maintain the structure stability in $V_2O_5$@rGO-x composites. Besides, the ultralow electron conductivity of 1.4×10$^{-5}$ S cm$^{-1}$ for pristine $V_2O_5$ can be significantly enhanced to 1.2 S cm$^{-1}$ for the $V_2O_5$@rGO-1.5 with the introduction of rGO (Supplementary Fig. 14). Thus, the coated rGO on $V_2O_5$ nanobelts can be acted as conductive layers in heterostructure to enhance the kinetics as well as satisfying power capability in energy conversion and storage.

## Evaluation of thermoelectrochemical performances

To correlate the structure regulation of $V_2O_5$@rGO-x with the enhancement of thermoelectrochemical performances in the harvesting of low-grade heat, the open-circuit voltage values of systems were recorded using non-isothermal H-type cell. Detailly, the modified Zn-G foil (thickness: ~20 μm) and $V_2O_5$@rGO-x are used as anode in cold side and cathode on hot side, respectively. It should be pointed out that the temperature difference (ΔT) between such two chambers can be formed by water bath and determined by inserted thermocouples. To evaluate the Seebeck coefficient of rGO, $V_2O_5$, $V_2O_5$@rGO-1.0, $V_2O_5$@rGO-1.5, and $V_2O_5$@rGO-2.0, we have profiled the related results with error bands or error bars (error band/bar represents the standard deviations). Benefitting to the adsorption/desorption mechanism of ions, the kinetics of $V_2O_5$ can be rationally enhanced with the addition of rGO. As summarized in Fig. 2a, the $V_2O_5$@rGO-1.5 based ZTCC generated the highest output voltage of ~1.1 V among all other ZTCCs. In fact, such high voltage can be divided as thermal-induced voltage and self-charging-induced voltage. The self-charging

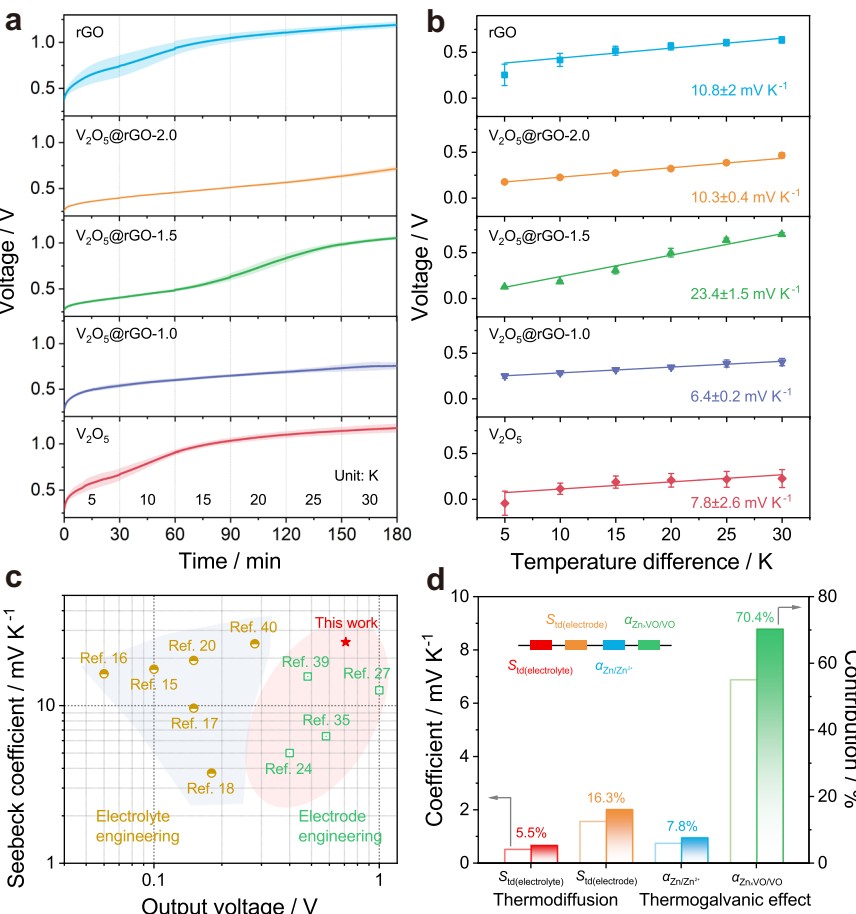

**Fig. 2 | Thermoelectrochemical performances of H-type ZTCCs using 0.5 mol L$^{-1}$ Zn(CF$_3$SO$_3$)$_2$. a** Output voltage under various temperature differences from 5 K to 30 K, and **b** corresponding Seebeck coefficient. **c** Comparison of Seebeck coefficient and thermal-induced voltage with other reported values. **d** Temperature coefficients and corresponding fractional contributions of various thermal processes.

behavior of Zn-related devices was carefully studied in previous works[38]. To exclude the contribution of the self-charge process to the total voltage, we further recorded the voltage curves during self-charging processes in Supplementary Fig. 15. Owing to the continuous and dense framework of rGO, the gradual desorption of ion from interconnected channels endows rGO-contained ZTCC with relatively low self-charging rate. Thus, the $V_2O_5$@rGO-1.5 based ZTCC delivers the highest thermal-induced voltage of 0.72 V. Based on these results, the thermopower or Seebeck coefficient ($S_i$) can be determined by the relationship between thermal-induced voltage and temperature difference[15,18]:

$$S_i = \frac{\Delta V}{\Delta T} \qquad (1)$$

As profiled in Fig. 2b, the rGO-based ZTCC shows large $S_i$ value of $10.8 \pm 2\,mV\,K^{-1}$ based on the ultrafast adsorption/desorption. While the $V_2O_5$ based ZTCC only delivers $7.8 \pm 2.6\,mV\,K^{-1}$ due to the relatively sluggish insertion/extraction of $Zn^{2+}$ in its crystalline layers. When combining the merits of rGO and $V_2O_5$, a giant $S_i$ value of $23.4 \pm 1.5\,mV\,K^{-1}$ can be rationally achieved by adjusting the content of rGO. In detail, the slight lower $S_i$ value of $V_2O_5$@rGO-1.0 based ZTCC than that of $V_2O_5$ based ZTCC may result from the its relatively dense structure and relatively low specific surface area. While for $V_2O_5$@rGO-2.0 based ZTCC, the $S_i$ value is close to that of rGO-based ZTCC, which may be caused by the stacking of as-introduced excessive rGO into $V_2O_5$. Accordingly, the Seebeck coefficient highly depends on the synergy behavior of thermal adsorption/desorption and thermal insertion/extraction. As is compared in Fig. 2c, this breakthrough is also superior to most of previously reported results[15–18,20,23,24,27,35,39,40], indicating the great promise of structure regulation by integrating graphene and vanadium oxides for high-efficient conversion from heat to electricity. As profiled in Supplementary Fig. 16, the rGO, $V_2O_5$, $V_2O_5$@rGO-1.0, $V_2O_5$@rGO-1.5, and $V_2O_5$@rGO-2.0 based ZTCC exhibits thermo-voltage of approximately 1.2, 1.2, 0.75, 1.1, and 0.7 V with the temperature difference of 30 K. It is worth mentioning that such output voltage can be further enhanced to 1.6 V by adding power charge. After discharging with a current density of $0.1\,A\,g^{-1}$, we can calculate that the ratio of thermal charge part in fully charged rGO, $V_2O_5$, $V_2O_5$@rGO-1.0, $V_2O_5$-1.5, and $V_2O_5$@rGO-2.0 based cell is 55.2%, 68.3%, 41.2%, 69.3%, and 30.0%, respectively. This result indicates that the heat-to-current conversion of ZTCC is highly determined by the kinetics of electrodes. Supplementary Fig. 17 displays the short-circuit current plots and fitted power density curves under thermal-induced voltage for various H-type ZTCCs. As found, the $V_2O_5$@rGO-1.5 based ZTCC delivers high voltage of 0.67 V and current density of $5.3\,A\,m^{-2}$ among rGO (0.39 V, $0.42\,A\,m^{-2}$), $V_2O_5$ (0.41 V, $0.59\,A\,m^{-2}$), $V_2O_5$@rGO-1.0 (0.59 V, $1.7\,A\,m^{-2}$), and $V_2O_5$@rGO-2.0 (0.59 V, $3.5\,A\,m^{-2}$). Further, a relatively low resistance value of $1118.7\,\Omega$ can be calculated from the $V_2O_5$@rGO-1.5 based ZTCC, while the value is 3071.3 and $1491.8\,\Omega$ for $V_2O_5$@rGO-1.0 and $V_2O_5$@rGO-2.0, respectively. Besides, the rGO based ZTCC delivers relatively high inner resistance of $8217\,\Omega$ due to its poor wettability for electrolyte. While the resistance of $6150\,\Omega$ for $V_2O_5$ based ZTCC is mainly caused by its ultralow electron conductivity. Meanwhile, the $V_2O_5$@rGO-1.5 based ZTCC exhibits an excellent thermal-induced power density of $0.94\,W\,m^{-2}$, meaning a high $P_{max}/(A \times \Delta T^2)$ value of $1.04\,mW\,m^{-2}\,K^{-2}$. Such values are greatly larger than that of $0.06\,W\,m^{-2}$ ($0.07\,mW\,m^{-2}\,K^{-2}$), $0.06\,W\,m^{-2}$ ($0.07\,mW\,m^{-2}\,K^{-2}$), $0.27\,W\,m^{-2}$ ($0.3\,mW\,m^{-2}\,K^{-2}$), and $0.65\,W\,m^{-2}$ ($0.72\,mW\,m^{-2}\,K^{-2}$) for rGO, $V_2O_5$, $V_2O_5$@rGO-1.0 and $V_2O_5$@rGO-2.0 based ZTCCs, respectively. As shown in Supplementary Fig. 18, the $V_2O_5$@rGO-1.5 based ZTCC maintains relatively stable voltage change over 6000 s

(60 cycles), demonstrating its good thermal charging stability. Due to the mismatched energy/kinetics during thermoelectrochemical process, the dramatic change in the voltage during the tests can be found in the rGO, $V_2O_5$, $V_2O_5$@rGO-1.0, and $V_2O_5$@rGO-2.0 based ZTCCs. Moreover, a negligible voltage drop (7 - 29 mV) occurs in self-discharge curves of various H-type ZTCCs when eliminating the temperature gradient (Supplementary Fig. 19). Especially, the $V_2O_5$@rGO-x based ZTCCs show relatively fast thermoelectrochemical response, and $V_2O_5$@rGO-1.5 based ZTCC delivers the lowest voltage change of ~7 mV, suggesting the dense energy storage behavior of as-proposed ZTCC. The gradual growth of voltage in following process can be attributed to the chemically self-charging behaviors[38]. To detailly study the relative contribution of thermo-diffusion and thermogalvanic process to such giant value, we conducted a series of measurements for various temperature coefficients from 25 to 50 °C using the three-electrode configuration proposed in previous works[17,35]. As briefly illustrated in Supplementary Fig. 20a, two same electrodes are employed as working electrode and counter electrode, respectively, and the Ag/AgCl is used as reference electrode. During tests, the temperature of setup is determined by hot water bath and recorded by thermocouple. As plotted in Supplementary Fig. 20b, the thermodiffusion of electrolyte ions can deliver a temperature coefficient of $0.54\,mV\,K^{-1}$. Notably, the temperature coefficient for $Zn/Zn^{2+}$ and $Zn_xVO/VO$ is 0.77 and $6.9\,mV\,K^{-1}$ (Supplementary Fig. 20c–f), respectively. Consequently, the total temperature coefficient ($9.8\,mV\,K^{-1}$) can be divided as 21.8% of thermodiffusion contribution and 78.2% of electronic effect contribution (Fig. 2d). All these findings reveal that the rational integration of capacitor-type rGO and battery-type $V_2O_5$ can realize the ultrafast response to heat/electricity signal.

## Kinetics analyses

The kinetics of as-obtained $V_2O_5$@rGO-1.5 cathode was reasonably studied by the cyclic voltammetry (CV) measurements and the galvanostatic intermittent titration techniques (GITT) with an operating voltage window from 0.2 to 1.6 V (Supplementary Note 1, Supplementary Figs. 21 and 22). As displayed in Fig. 3a, two pair of visible redox peaks at 0.4/0.7 V and 0.9/1.1 V in all the CV curves from 0.1 to 1.0 mV s$^{-1}$ can be attributed to the multiple insertion/extraction procedures of $Zn^{2+}$ in $V_2O_5$ ($V_2O_5 + xZn^{2+} + yH_2O + 2xe^- \leftrightarrow Zn_xV_2O_5 \cdot yH_2O$). The well-maintained CV curves even at relatively high scan rate further suggest the good reversibility and rapid response of $V_2O_5$@rGO-1.5 electrode. As one of convenient indicators, the near surface-dominated procedure and bulk diffusion-controlled process can be distinguished by using $i = av^b$, where $i$ and $v$ represent the peak current and scan rate, $a$ and $b$ are adjustable parameters[41]. Typically, $b = 0.5$ means that the process is totally governed by diffusion, while $b = 1.0$ suggests the capacitive behavior. As plotted and fitted in Fig. 3b, the as-calculated $b$ value for peaks 1–4 are 0.78, 0.86, 0.79, and 0.81, respectively, revealing that the electrochemical process of the $V_2O_5$@rGO-1.5 is monopolize-dominated by the capacitive process along with partial diffusion contribution. Moreover, the detail contribution of such two processes can be clarified by the following relationship:[42]

$$i = i_{cap} + i_{diff} = k_1 v + k_2 v^{1/2} \qquad (2)$$

As specifically shown in Supplementary Fig. 12, a high capacitive contribution of 80.8% can be achieved at the scan rate of 1.0 mV s$^{-1}$. Moreover, the contribution ratio of the capacitive process shows a gradual increase trend from 0.1 to 1.0 mV s$^{-1}$ (Fig. 3c). The capacitance dominated process in each scan rate is in line with the result discussed above. Meanwhile, the GITT was used to investigate the $Zn^{2+}$ diffusion coefficient in electrodes to highlight the structural merits of $V_2O_5$@rGO-1.5 nanocomposite in Zn-based

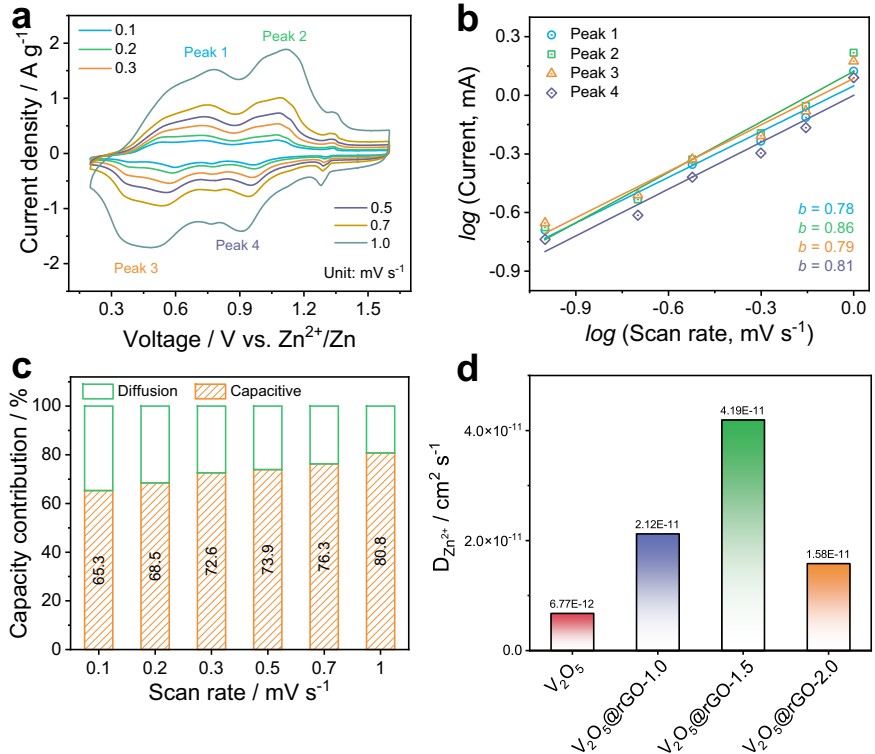

**Fig. 3 | Electrochemical kinetics analysis of $V_2O_5$@rGO-1.5. a** CV curves at different scan rates. **b** $log$ (Current) *vs. log* (Scan rate) of various peaks, **c** the capacity contribution proportion at various scan rates. **d** $Zn^{2+}$ diffusion coefficient obtained by GITT curves at charging/discharging.

thermoelectrochemical devices. The corresponding GITT curves during charging and discharging processes are displayed in Supplementary Fig. 23. Accordingly, the $Zn^{2+}$ diffusion coefficient ($D_{Zn}$) of $V_2O_5$ is in a range of $10^{-12}$ to $10^{-11}$ $cm^2 s^{-1}$ during the whole electrochemical tests (Supplementary Fig. 23a, e), demonstrating relatively sluggish insertion/extraction of $Zn^{2+}$ in the pristine $V_2O_5$. With the increase of rGO content, the $D_{Zn}$ value can be enhanced to a range of $10^{-11}$ to $10^{-10}$ $cm^2 s^{-1}$ for $V_2O_5$@rGO-1.0 and $V_2O_5$@rGO-1.5 (Supplementary Fig. 23b, f and c, g), which is almost an order of magnitude higher than the value of $V_2O_5$. However, the $D_{Zn}$ value shows a slight loss in the $V_2O_5$@rGO-2.0 (Supplementary Fig. 23d and h), which mainly caused by the dense wrapping of $VO_x$ species by stacked rGO layers. As summarized in Fig. 3d, $V_2O_5$@rGO-1.5 exhibits the highest average $Zn^{2+}$ diffusion coefficient of $4.19×10^{-11}$ $cm^2 s^{-1}$ during the whole electrochemical processes. Such relatively fast insertion/extraction of $Zn^{2+}$ in the $V_2O_5$@rGO-1.5 electrode may be owing to the relatively large interlayer spacing of $V_2O_5·1.6H_2O$ and the high conductivity of rGO substrate. Most importantly, the crystal water may not only effectively reduce energy barrier but also mask the electrostatic interaction of $Zn^{2+}$ via solvation effect[43–45]. The slightly low $D_{Zn}$ value of charging part implies the sluggish extraction kinetics of $Zn^{2+}$ from cathode, which requires high energy input to realize such charging procedure. Based on this feature, high efficiency and density can be achieved for the conversion from low-grade heat to electricity by using $V_2O_5$@rGO-1.5 based ZTCCs.

**Energy conversion and storage behaviors**

As attractive system integrated energy conversion and storage, the zinc ion batteries (ZIBs) are assembled to evaluate the electrochemical performances. Notably, the $V_2O_5$@rGO-1.5 based ZIBs shows superior rate capability (Supplementary Fig. 24a), in which 375.5, 383.5, 376.9, 353.4, 320.4, 266.3, 208.0, and 145.0 mAh $g^{-1}$ can be recorded at 0.1, 0.2, 0.5, 1.0, 2.0, 5.0, 10.0, and 20.0 A $g^{-1}$, respectively. Impressively, the reversible discharging capacity can be nearly stabilized at

402.8 mAh $g^{-1}$ as the current density recovers to 0.1 A $g^{-1}$. The slight increase of specific capacity after test could be caused by the electrolyte penetration induced activation process[35]. The rate performance of $V_2O_5$@rGO-1.5 cathode is also better than that of rGO, $V_2O_5$, $V_2O_5$@rGO-1.0, $V_2O_5$@rGO-2.0 electrodes and some of previously reposted materials (Supplementary Fig. 25a), such as $VO_2$-PC (82.6 mAh $g^{-1}$ at 20 A $g^{-1}$)[27], VPMX73 (282.2 mAh $g^{-1}$ at 3 A $g^{-1}$)[46], ZVO (167 mAh $g^{-1}$ at 15 A $g^{-1}$)[47], $Zn_{0.25}V_2O_5·nH_2O$ (183 mAh $g^{-1}$ at 6 A $g^{-1}$)[43], MnVO (214 mAh $g^{-1}$ at 8 A $g^{-1}$)[31], $Ni_{0.25}V_2O_5·nH_2O$ (164 mAh $g^{-1}$ at 5 A $g^{-1}$)[48], NVO/CNTs (203 mAh $g^{-1}$ at 4 A $g^{-1}$)[49], NaCaVO (154 mAh $g^{-1}$ at 5 A $g^{-1}$)[45], and $V_2O_5·1.6H_2O$/MXene (81.2 mAh $g^{-1}$ at 2 A $g^{-1}$)[50]. As plotted in Supplementary Fig. 25b, the $V_2O_5$@rGO-1.5 based devices exhibits the highest energy density of 265 Wh $kg^{-1}$ at a power density of 151 W $kg^{-1}$ based on the active mass loading of cathode. Even under the highest power density of 12964.3 W $kg^{-1}$, a high energy density of 84.7 Wh $kg^{-1}$ still can be retained, suggesting the superior energy storage behavior and rate capability of $V_2O_5$@rGO-1.5 among other electrode materials. Supplementary Fig. 24b displays the specific capacity retention of various electrodes at the current density of 10 A $g^{-1}$. Initially, the capacity of $V_2O_5$@rGO-1.5 increases to 195 mAh $g^{-1}$ due to the gradual activation. Such value can maintain about 122 mAh $g^{-1}$ even over 5000 cycles together with the corresponding Coulombic efficiency of ~100%, suggesting satisfying long cyclic stability. Notably, an obvious increase of capacity before 2000 cycles can be found in $V_2O_5$ cathode, this may be caused by the electrochemical activation process. However, large capacity decay of $V_2O_5$ cathode in following cycles may be caused by the dissolution of vanadium. Similar phenomenon can be found in $V_2O_5$@rGO-1.0 cathode. In addition, the cycled $V_2O_5$ electrode shows a distinct morphology change due to the obvious vanadium dissolution (Supplementary Fig. 26a). The cycled rGO electrode still holds the inter-connected morphology with abundant pores (Supplementary Fig. 26b), in line with its initial morphology. In sharp contrast, the cycled $V_2O_5$@rGO-x electrodes display similar structure and morphology to their pristine state (Supplementary Fig. 26c–e), and rGO

layers can be visually observed in each electrode. From the EDX mapping images of each electrode after cycling, we can find that the zinc species exhibit in all electrodes due to the irreversible electrochemical reactions. Notably, both V$_2$O$_5$@rGO-2.0 and V$_2$O$_5$@rGO-1.5 electrodes present the relatively uniform distribution of C, O, Zn, and V elements (Supplementary Fig. 26d, e), demonstrating the effect of rGO to suppress the vanadium dissolution. However, the distribution of C and V in cycled V$_2$O$_5$@rGO-1.0 electrode is relatively uneven, which may be caused by the dissolution of vanadium. Therefore, both the V$_2$O$_5$@rGO-2.0 and V$_2$O$_5$@rGO-1.5 electrodes can achieve good stability when comparing with the V$_2$O$_5$@rGO-1.0 electrode. As plotted in Supplementary Fig. 27, the Seebeck coefficient value of each system by re-using cycled electrodes is relatively lower than that using fresh working electrodes. Especially, the ultralow Seebeck coefficient of $0.75 \pm 0.1$ mV K$^{-1}$ of pristine V$_2$O$_5$ is mainly caused by the serious dissolution of vanadium species during long-term cycling tests. However, the ZTCC with cycled V$_2$O$_5$@rGO-1.5 still can deliver a relatively high Seebeck coefficient of $17.5 \pm 2.6$ mV K$^{-1}$, further demonstrating that the rGO coating on V$_2$O$_5$ nanobelt shows key role to achieve satisfying durability and stability of vanadium-based devices by suppressing the dissolution of vanadium species to electrolyte. It should point out that the slight decrease of Seebeck coefficient value for V$_2$O$_5$@rGO-1.0, V$_2$O$_5$@rGO-1.5, V$_2$O$_5$@rGO-2.0, and rGO based ZTCCs could be the influence of byproducts.

Besides, various temperatures from 30 °C to 50 °C were adopted to further investigate the merits of heterostructure. With the increase of testing temperature (30 °C and 40 °C), the specific capacity of all cathodes can be enhanced and the electrochemical activation process can be accelerated due to the thoroughly infiltration of electrolytes (Supplementary Fig. 24c, d). It is worth mentioning that the V$_2$O$_5$@rGO-2.0 cathode exhibits better rate capability and cycling stability than other electrodes at 40 °C (Supplementary Fig. 24e, f), which is mainly attributed to its lowest dissolution rate of vanadium into electrolyte among other materials. Apart from the dissolution of vanadium, the dendrite growth and byproduct formation on Zn-G anode under relatively high temperatures (50 °C) would become the culprit of cell failure. Worthily, all as-assembled cells will be short-circuited when the total testing time is around 80 h (Supplementary Fig. 24g, h). Even vanadium dissolution at high temperatures will intensify, the cells should be operated with relatively low capacity rather than short-circuited. Thus, the biggest issue of Zn-based cells under high temperatures could be the serious growth of dendrite.

Furthermore, only a low absolute temperature coefficient of 2.5 mV K$^{-1}$ can be obtained from the CV curves of as-developed V$_2$O$_5$@rGO-1.5 based ZTCC, which is much lower than the Seebeck coefficient ($23.4 \pm 1.5$ mV K$^{-1}$). Such relatively small polarization potential (~50 mV) indicates the fast electron transfer kinetics and the better reversibility of ZTCCs. Noted from equation S7 in Supplementary Note 2, in addition to the tested temperature coefficient, some possible factors also could affect thermopower or Seebeck coefficient of ZTCCs. The first one is related to the entropy change of redox species. The second one is the concentration difference of redox species in ZTCCs, which greatly impacts the whole Seebeck coefficient due to the possible chemical reactions between the two electrodes. The last one is the thermodiffusion of electrolyte ions in ZTCCs. Meanwhile, the CV curves of both H-type ZTCCs under different temperature differences and coin-type cells under various temperatures indicate the polarization of redox reactions between V$_2$O$_5$ and Zn can be significantly optimized by adding moderate rGO (Supplementary Note 2, Supplementary Figs. 28 and 29).

To well understand the energy conversion and storage mechanism of ZTCC, we observed the morphology change of both V$_2$O$_5$@rGO-1.5 cathode and Zn-G anode during the thermoelectrochemical measurements by SEM. As shown in Supplementary

Fig. 30a1, the V$_2$O$_5$@rGO-1.5 is evenly wrapped by carbon black and PVDF in the initial electrode. After discharging to 0.2 V, some thin films were covered on the surface of the electrode along with the Zn$^{2+}$ insertion to the V$_2$O$_5$@rGO-1.5 cathode (Supplementary Fig. 30a2), which may be caused by the generation of by-product like Zn$_4$(CF$_3$SO$_3$)$_4$(OH)$_4$·3H$_2$O. Notably, as-formed film can be gradually decomposed with the increase of temperature difference from 5 to 30 K (Supplementary Fig. 30a3–a8). Moreover, the graphite layers are covered on the surface Zn-G (Supplementary Fig. 30b1), implying the successful modification of pure Zn anode[37]. Even discharged to 0.2 V (Zn$^{2+}$ ions are stripped from the Zn-G anode), the graphite layers still can be maintained on the surface of Zn-G (Supplementary Fig. 30b2). Worthily, the morphology of Zn-G anode during the whole thermoelectrochemical process changes obviously. When the temperature difference is 5 K (Supplementary Fig. 30b3), some byproducts nucleate on the surface of Zn-G anode with the plating of Zn$^{2+}$. Furthermore, the Zn-G anode displays a well-defined array structure formed by numerous thin nanosheets (Supplementary Fig. 30b4–b8). We also conducted structural characterizations of V$_2$O$_5$@rGO-1.5 cathode during various charge/discharge states with ex-situ XRD and XPS analyses to investigate such processes. Specifically, the V$_2$O$_5$@rGO-1.5 electrode at different voltages is marked as adopted temperature differences (x K) and state y, as depicted in Fig. 4a. It should point out that the sharp peaks at ~26.4° and 54.5° in all XRD patterns correspond to the graphite substrate (Fig. 4b and c). Compared to the initial electrode, the pattern of the electrode at 0.2 V after the first discharge shows the obvious right shift of characteristic peak, suggesting the transformation of the electrode from V$_2$O$_5$·1.6H$_2$O to Zn$_x$V$_2$O$_5$·nH$_2$O due to the co-insertion of Zn$^{2+}$ and H$_2$O[46]. During the thermal charge process, the (001) plane of Zn$_x$V$_2$O$_5$·nH$_2$O gradually shifts to a low-angle region and shows a tendency to change back to V$_2$O$_5$·1.6H$_2$O (Fig. 4b), indicating the thermoextraction of Zn$^{2+}$/H$_2$O from crystals. Moreover, the formed Zn$_x$V$_2$O$_5$·nH$_2$O almost fully recovers as V$_2$O$_5$·1.6H$_2$O (state I) after the following electrochemically charging process to 1.6 V (Fig. 4c). Notably, the reversible shift of (001) plane from state I to state V well confirms the highly reversible conversion between Zn$_x$V$_2$O$_5$·nH$_2$O and V$_2$O$_5$·1.6H$_2$O, demonstrating high reversibility for energy storage. This phenomenon also suggests the co-insertion of Zn$^{2+}$ and H$_2$O into V$_2$O$_5$·1.6H$_2$O during the discharging process, which then adjusts back to V$_2$O$_5$·1.6H$_2$O owing to the extraction of ions during the following charging process. Notably, the weak peak around 17.1° during the thermoelectrochemical process can be attributed to the presence of Zn$_4$(CF$_3$SO$_3$)$_4$(OH)$_4$·3H$_2$O, confirming the formation of byproduct on electrode surface[42,51,52].

The chemical states and electronic interaction during fully charge and discharge stages are further investigated by using XPS measurements (Fig. 4d–f). As shown in Fig. 4d, two strong peaks located at 1022 and 1045 eV from state III can be assigned to the Zn $2p_{3/2}$ and Zn $2p_{1/2}$ signals, which is reasonably caused by the insertion of Zn$^{2+}$ into the vanadium oxide layers. After charging to state V, the Zn $2p$ signals with lower intensity indicate that a lot of Zn$^{2+}$ can be reversibly extracted, while partial residual zinc species still present in V$_2$O$_5$@rGO-1.5 cathode[35]. When considering the valence change of V element during electrochemical processes, the electrode at state III shows a strong V$^{4+}$ peak together with a high V$^{4+}$/V$^{5+}$ ratio of 1.06, meaning a significant reduction of V$_2$O$_5$ with the insertion of Zn$^{2+}$ (Fig. 4e). Upon recharging to state V, the V$^{5+}$ peak becomes strong and the V$^{4+}$/V$^{5+}$ ratio (0.31) is almost recovered to the initial state (0.23), in consistent with the extraction of Zn$^{2+}$. In addition, the content of H$_2$O in O $1s$ spectra exhibits significant change after discharge and charge (Fig. 4f), confirming the solvation effect endowed insertion of H$_2$O associated with the major insertion of Zn$^{2+}$[43]. According to the above results, the possible electrochemical reactions can be proposed as follows:

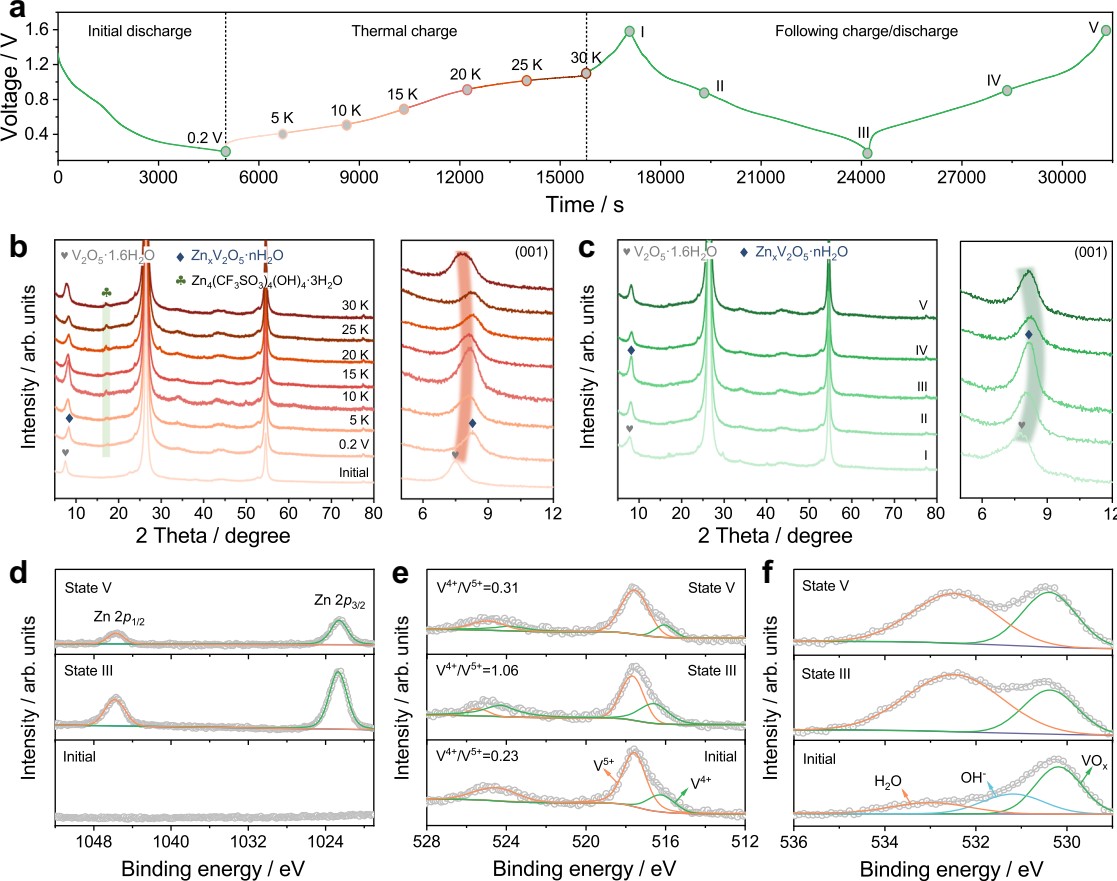

**Fig. 4 | The mechanism analyses of V₂O₅@rGO-1.5 based ZTCCs. a** The voltage change curves at different processes. **b, c** The corresponding ex-situ XRD patterns of V₂O₅@rGO-1.5 cathode during **b** thermal charging states at various temperature differences and **c** electrochemically charging/discharging states at 30 K. **d−f** The corresponding ex-situ XPS spectra of **d** Zn 2p, **e** V 2p, and **f** O 1s.

**Cathode:**

$$V_2O_5\cdot1.6H_2O + xZn^{2+} + 4H^+ + (n-1.6)H_2O + 2xe^- \leftrightarrow H_4Zn_xV_2O_5\cdot nH_2O \quad (3)$$

$$4H_2O \leftrightarrow 4H^+ + 4OH^- \quad (4)$$

$$2Zn^{2+} + 2Zn(CF_3SO_3)_2 + 4OH^- + 3H_2O \leftrightarrow Zn_4(CF_3SO_3)_4(OH)_4\cdot3H_2O \quad (5)$$

**Anode:**

$$xZn \leftrightarrow xZn^{2+} + 2xe^- \quad (6)$$

## Demonstration of wearable devices

Quasi-solid-state devices shows great promise in multi-functional applications due to their flexibility, safety, and wearability[53]. Here, the wearable ZTCC was using polyacrylamide (PAM) based gel electrolyte, which is sandwiched by Zn-G anode and V₂O₅@rGO-1.5 cathode, as illustrated in Supplementary Fig. 31. It is worth mentioning that the temperature difference between two electrodes is generated by a resistive heater (12 V, 7 W) attached on the cathode side. The as-assembled device delivers the output voltage of 0.38 V at an ultralow temperature difference of 8 K among other solid-state devices (Fig. 5a). After deducting the self-charging contribution (Supplementary Fig. 32), the thermal-induced voltage is 55, 74, 89, 101, 111, 117, 126, and 122 mV at the temperature difference from 1 to 8 K, respectively. As summarized

in Fig. 5b, the corresponding Seebeck coefficient of V₂O₅@rGO-1.5 based solid-state device is calculated to be around $11.9 \pm 1\,mV\,K^{-1}$, reflecting good thermal-electrical response in near room temperature low-grade heat harvesting. Such value also is much higher than that of rGO ($10.4 \pm 1\,mV\,K^{-1}$), V₂O₅ ($8.9 \pm 1\,mV\,K^{-1}$), V₂O₅@rGO-1.0 ($5.2 \pm 0.7\,mV\,K^{-1}$), and V₂O₅@rGO-2.0 ($9.4 \pm 0.7\,mV\,K^{-1}$) based devices. Remarkably, the highest thermal-induced power density of $0.12\,W\,m^{-2}$ can be obtained by V₂O₅@rGO-1.5 among all as-fabricated solid-state devices (Fig. 5c), signifying an ultrahigh normalized power density of $1.9\,mW\,m^{-2}\,K^{-2}$ in state-of-the-art reported systems (Supplementary Table 1). Inspired by such impressive performances of V₂O₅@rGO-1.5 based solid-state ZTCC, an external load with a resistance of 10 kΩ was employed to examine its stability and durability in energy conversion. As displayed in Fig. 5d, the output voltage was stably maintained at ~0.4 V with the adoption of temperature gradient, implying the satisfying thermal stability and promise potential of solid-state ZTCC. However, the relatively high voltage decay of rGO and V₂O₅ based solid-state ZTCC are possibly caused by the poor capacity and sluggish ion diffusion kinetics, respectively. In contrast, the slight voltage changes of V₂O₅@rGO-1.0 and V₂O₅@rGO-2.0 based solid-state ZTCC could be attributed to their relatively low ion diffusion coefficient. Even after 7 days, the V₂O₅@rGO-1.5 based solid-state ZTCC still can realize the thermal charge and discharge processes (Supplementary Fig. 33). This result demonstrates that as-developed ZTCCs can be used repetitively for relatively long-term cycling rather than being a one-time energy source. As a demo, three ZTCCs connected in series can easily light up a white light-emitting diode (LED) with a temperature difference of ~10 K (Fig. 5e). When considering the temperature difference between body

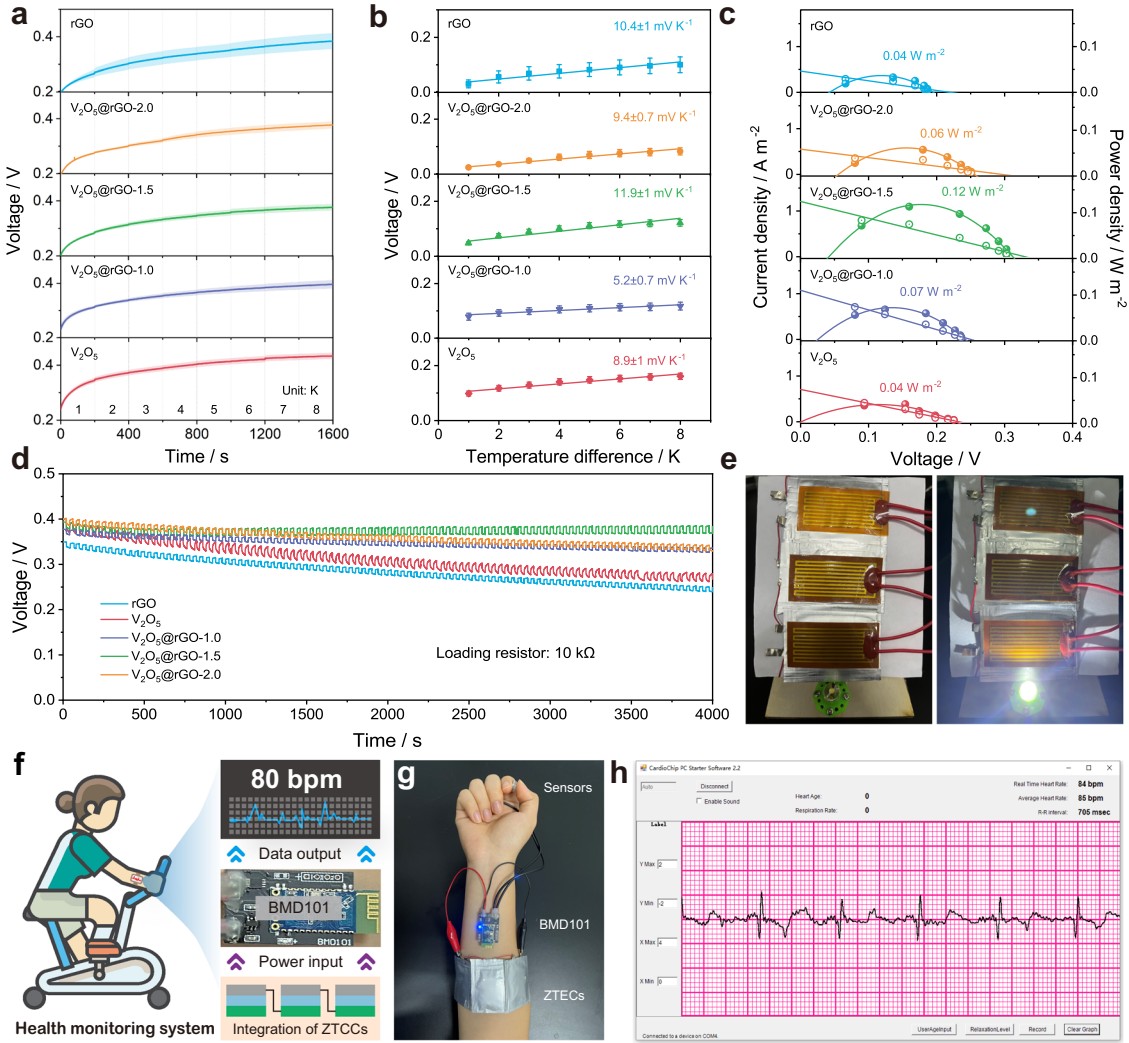

**Fig. 5 | Construction of quasi-solid-state ZTCCs using PAM based gel electrolyte and applications. a** Thermal charging curves under various temperature differences. **b** Fitted Seebeck coefficient. **c** Thermal-induced current density (hollow symbol) and power density (solid symbol). **d** Thermal stability with a loading resistance of 10 kΩ. **e** Digital photo of a white LED lighted by three ZTCCs in series. **f** Illustration for health monitoring system. **g** Body heat-charged health monitoring system and **h** the recorded data.

heat and ambient, $V_2O_5$@rGO-1.5 based solid-state ZTCC is very attractive as one of power input source to replace traditional batteries for integrated health monitoring systems (Fig. 5f). Figure 5g shows the digital photo of a brief health monitoring system. Typically, the selected electro-cadiography sensor module (BMD101) can be powered by three ZTCCs connected in series. By Bluetooth wireless, the recorded data can be presented and processed in the mobile phone or laptop (Fig. 5h). All above-mentioned demonstrations further confirm the application prospect of as-constructed ZTCCs in energy conversion and storage fields, which makes it possible in terms of electronics integration.

## Discussion

In summary, this work has developed high-performance ZTCCs by cathode engineering. By sequentially introducing 2D rGO into 1D $V_2O_5 \cdot 1.6H_2O$ nanobelts, both the thermal response and kinetics of as-constructed ZTCCs are modified. Notably, the $V_2O_5 \cdot 1.6H_2O$ nanobelts provide enough electroactive sites for dense $Zn^{2+}$ storage, suggesting high capacity and energy density. The rGO layers wrapped around $V_2O_5 \cdot 1.6H_2O$ nanobelts can not only keep structural stability and inhibit V dissolution, but also accelerate the charge transfer, implying good durability and excellent power density. As a result, a giant thermopower of $23.4 \pm 1.5$ mV K$^{-1}$ can be achieved by $V_2O_5$@rGO-1.5 based

ZTCC, which synergistically integrates the fast thermodiffusion and thermoextraction processes. Moreover, a high thermal-induced voltage of 0.72 V and normalized power density of 1.04 mW m$^{-2}$ K$^{-2}$ can be delivered at the temperature difference of 30 K. The ex-situ experiments and DFT calculations reveal that the rGO can boost the charge transfer at the heterointerfaces formed by rGO and $V_2O_5 \cdot 1.6H_2O$. Besides, the crystal water layer in $V_2O_5 \cdot 1.6H_2O$ can promote the diffusion of $Zn^{2+}$ with low energy barrier due to its reduced electrostatic repulsion effect. It is worth mentioning that as-proposed $V_2O_5$@rGO-1.5 based ZTCC shows great promise in self-power supply for health monitoring systems. By coupling with hydrogel electrolyte, three solid-state ZTCCs attached to the arm can drive the Bluetooth wireless module by an ultralow temperature difference formed between body heat and atmosphere, demonstrating potential application of ZTCCs in energy conversion and storage as well as wearable electronics areas.

## Methods

### Preparation of $V_2O_5$@rGO-x

To prepare $V_2O_5$@rGO-x, the concentration of commercial graphene oxide (GO, CARMERY, Institute of Coal Chemistry, Chinese Academy of Sciences) solution was firstly adjusted to 1.0, 1.5, and 2.0 mg mL$^{-1}$. Then, 120 mg of vanadium pentoxide ($V_2O_5$, Xiya Reagent) was

dissolved in 60 mL of GO solution and stirred magnetically for about 30 min at room temperature. Subsequently, above solution was transferred into a 100 mL Teflon-lined stainless steel autoclave and kept at 180 °C for 12 h. Finally, the products were collected by centrifugation at 9166 RCF xg for 3 min and washed several times using deionized water and ethanol, followed by drying overnight. The samples are labeled as $V_2O_5$@rGO-x, where x represents the GO concentration. For comparison, we carried out control experiments with pure GO solution (1.5 mg mL$^{-1}$) and $V_2O_5$ in water by the same conditions to obtain rGO and $V_2O_5$, respectively.

## Material characterizations

The morphology was observed by using a field emission scanning electron microscope (FESEM, FEI NANO SEM430) and a high-resolution transmission electron microscope (HRTEM, JEOL JEM-2100). X-ray diffraction (XRD) patterns were recorded from PANalytical Empyrean diffractometer with Cu Kα radiation (λ = 1.5406 Å). The surface chemical states of samples were detected by X-ray photoelectron spectrometer (XPS, Thermo Scientific K-Alpha). The Raman spectra were collected by the Horiba Scientific LabRAM HR with an excitation wavelength of 532 nm. Thermogravimetric analysis (TGA) curve was recorded on a Mettler-Toledo STARe SW 15.00 analyzer with a heating rate of 10 °C min$^{-1}$ under air flow. The wettability of materials was measured by a CA tester (XG-CAMC33) using electrolyte drop (10 μL). The dissolution rate of vanadium into electrolyte was detected by Inductively Coupled Plasma Optical Emission spectroscopy (ICP-OES, Thermo Fisher iCAP PRO). The SSA of samples were measured by the ASAP 2460 analyzer (micromeritics), and were analyzed by Brunauer-Emmett-Teller (BET) method.

## Electrochemical measurements

Thermoelectrochemical performance tests of ZTCCs were conducted in a non-isothermal H-type cell with 0.5 mol L$^{-1}$ Zn(CF$_3$SO$_3$)$_2$ electrolyte on a standard electrochemical workstation (CHI 760E). Zn-G foil (thickness: ~20 μm) and as-prepared electrodes were used as counter/reference electrodes and working electrodes, respectively. The Zn-G anode was modified by graphite using the pencil drawing method. To prepare the working electrode, $V_2O_5$@rGO-x, acetylene black, and polyvinylidene fluoride were firstly mixed together in a mass ratio of 7:2:1. Then, the slurry was painted on graphite paper with a diameter of 1.2 cm (mass loading: 1.2 mg cm$^{-2}$) as working electrode. CV curves of ZIBs were recorded from 0.1 to 1.0 mV s$^{-1}$ with a voltage window of 0.2–1.6 V by the Biologic VMP-300 workstation. GITT, rate capability, and cyclic stability were measured using the CT3001A Land Battery Test System.

The quasi-solid-state ZTCC was assembled by Zn-G foil anode and $V_2O_5$@rGO-x cathode together with hydrogel electrolyte. It should mention that both the Zn-G and $V_2O_5$@rGO-x electrodes hold the area of 2×3 cm$^2$, and the active mass loading in the $V_2O_5$@rGO-x electrode is about 8 mg. In addition, the PAM gel electrolyte was prepared by following steps. 2 g of acrylamide (AM, Macklin) and 0.4 g of acrylic acid (AA, Macklin) were dispersed into 10 mL mixed solvent of deionized water and glycerol in a volume ratio of 1:2. Subsequently, 1.5 mg of N,N'-methylenebisacrylamide (Macklin), 10 μL of N,N,N',N'-tetramethylethylenediamine (Aladdin), and 0.25 g of potassium persulfate (Aladdin) were added into above solution part by part with vigorous stirring for 4 h under ice-water bath. After that, the transparent solution was injected in a mold and polymerized by using a UV lamp (365 nm, 60 W) for 20 min to prepare the hydrogel matrix. Moreover, such hydrogel soaked with 0.5 mol L$^{-1}$ Zn(CF$_3$SO$_3$)$_2$ electrolytes can be used for the construction of quasi-solid-state ZTCCs.

## Data availability

All relevant data that support the findings of this study are presented in the manuscript and supplementary information file. Source data are available from the corresponding author upon reasonable request.

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

## Acknowledgements

This work was supported by the Leading Edge Technology of Jiangsu Province (BK20220009-X.Z., BK20202008-X.Z.) and Priority Academic Program Development of Jiangsu Higher Education Institutions (PAPD). We thank Qian Zhu, Xiuwen Wang, Zhenxiao Ling (Nanjing University of Aeronautics and Astronautics) for their kind help. We thank the Center for Microscopy and Analysis at the Nanjing University of Aeronautics and Astronautics for the advanced facilities.

## Author contributions

Z.L. carried out the experiment. Z.L., Y.X. and J.C. evaluated the electrochemical performances. L.W. analyzed the structural characterization and electrochemical results. Z.L. and Y.X. organized the figures. H.D. and X.Z. supervised the whole work. All authors co-discussed the results and wrote the manuscript.

## Competing interests

The authors declare no competing interests.
