## [Peer Review File · Nature Communications]

Enabling giant thermopower by heterostructure engineering of hydrated vanadium pentoxide for zinc ion thermal charging cellsREVIEWER COMMENTS

Reviewer #1 (Remarks to the Author):

The authors contribute a nice work of zinc ion thermoelectrochemical cells with giant thermopower for low-grade heat harvesting. Based on the thermoextraction effect and thermodiffusion effect, a record-level thermopower of 25.3 mV K⁻¹ and an excellent normalized power density of 2.7 mW m⁻² K⁻² were achieved. The great performances and the mechanism analyses of V₂O₅@rGO-1.5 based ZTECs were demonstrated, and the potential in the application of self-powered electronics was impressive. Moreover, as-constructed ZTECs displayed good energy storage behavior. In my view, such findings in this work are significant and provide a new strategy to fabricate high-performance ionic thermoelectrochemical devices by reasonable electrode engineering. It can be recommended for publication in Nature Communications. However, some specific comments listed below should be further discussed.

1. For ionic thermoelectrics, the stability of the measuring the thermopower/Seebeck coefficient is very important. The number of parallel experiments or error bars should be provided.
2. In the ZTECs, redox reaction occurs on the electrode during thermal charging. How does the electron transfer in the open circuit?
3. To clarify the energy conversion mechanism, the morphology change of both cathode and anode under various temperature differences should be provided.
4. How about the energy density of constructed ZTECs? Some related results and contents should be added.
5. What is the ratio of heat charge part in a fully charged battery with hybrid method?
6. Besides, can the voltage of ZTECs be maintained continuously if the temperature gradient is eliminated?

Reviewer #2 (Remarks to the Author):

The manuscript by Li et al. reports a zinc ion thermoelectrochemical cell using the hydrated vanadium pentoxide/rGO (V₂O₅@rGO) as the cathode and Zn/graphene as the anode, respectively. The basic concept of the thermoelectrochemical cell has already been reported, and a compositionally very similar cathode (i.e., VO₂-PC) has also been explored. Thought the current work shows enhanced performance, it does not provide new insights nor unexpected results. In addition, the detailed mechanisms are not investigated thoroughly. Many phenomena are not explained. For example, the charging/discharging mechanisms of V₂O₅ cathodes with different concentrations of rGO is not explored in-depth. The relationship between the Seebeck coefficient of the V₂O₅@rGO and the performance (capacity or lifespan) of the cathodes is unclear. I therefore do not recommend the publication of this manuscript.

1. The novelty of this work should be clearly demonstrated. What are the advantages of V₂O₅ against the VO₂ or other vanadium-based materials should be discussed.
2. The structure characterization is very limited. The authors have compared the thermoelectrochemical performance of the V₂O₅@rGO-1.0, V₂O₅@rGO-1.5, and V₂O₅@rGO-2.0, however, the origins account for such disparity are not investigated. Authors only roughly compared the morphology of these cathodes, which provide limited information. Whether these materials possess different electronic structure, coordination

environmental, surface area are not clear. It is therefore difficult to understand the relationship between the cathode composition/structure and the performance.

3. More evidences of the heterostructure should be provided.

4. The authors have shown that the output voltages of the various cathodes are significantly different. Do the different contents of rGO affect the thermo adsorption and the ion diffusion? Please give an explanation.

5. The authors mentioned that the use of rGO could enhance the stability of V₂O₅ by suppressing the metal dissolution. However, there are not supporting data. They should at least compare the performance of V₂O₅ and rGO and further monitor the metal dissolution rate. Same question goes to the Zn@G anode.

6. Will the Seebeck coefficient of V₂O₅@rGO-1.5 and the thermo-diffusion/extraction of ions in cathode be affected during cycling at different voltages, given the generation of new phases (e.g., Zn_xV₂O₅) and byproducts?

7. The Seebeck coefficient of V₂O₅ is significantly enhanced by the rGO. Therefore, the stability and distribution of rGO in the V₂O₅@rGO cathodes, especially after cycling, should be further investigated. Additionally, the Seebeck coefficient of V₂O₅@rGO-1.5 (25.3 mV K⁻¹) is close to that rGO (22.4 mV K⁻¹), whereas V₂O₅@rGO-1.0 and V₂O₅@rGO-2.0 exhibit significantly lower Seebeck coefficient than rGO, why?

8. The author should provide more cyclic stability and capacity data of the cells with rGO, V₂O₅, V₂O₅@rGO-1.0, V₂O₅@rGO-1.5, and V₂O₅@rGO-2.0 cathodes that measured under various temperature differences besides the only one long cycling performance at the room temperature.

9. The PAM gel is one of excellent materials for sensors, which exhibits high sensitivity for the temperature. Therefore, it is not surprisingly to see the trend shown in Fig. 5b-h. The experimental data of devices with rGO, V₂O₅, V₂O₅@rGO-1.0, and V₂O₅@rGO-2.0 cathodes should be also provided for comparison to confirm the enhanced performance of the V₂O₅@rGO-1.5 cathode.

Reviewer #3 (Remarks to the Author):

The authors reported exceptionally high thermoelectric power using the mixture of V₂O₅ and rGO. The demonstration seems interesting. However, the manuscript lacks a basic understanding of thermo-electrochemical cells, and the manuscript seems unable to be published.

The fundamental point is that the cell consists of redox-active electrodes, not electrolytes; thus, this is no more TEC. The electric cycle of TEC is achieved by the diffusion of redox-active species (both anolyte and catholyte). Thus, redox-active electrodes are interesting but not suitable for TECs. In other words, the observed electric power could be derived from electrochemical reactions between rGO and V₂O₅. The authors should show the total cycle of the cell.

Second, such high "thermoelectric voltage" could be derived from overpotentials. The overpotential or mixed potential of electrode materials is sometimes strongly affected by temperature. Thus, cyclic voltammetry or repeated potential measurements should be

executed to accurately estimate the effects of these voltages as well as equilibrium potentials. The temperature-dependent CV should be revealed. The repeated thermal charge measurement in Fig. 4a could give important information.

The followings are other trivial points to be revised.

1. Figure captions should include more information. Measurement temperature and detailed information on the electrode or solution should be added. The mass of electrodes is important to discuss the relation between the total redox charge and the amount of active materials.
2. The resolution of EDX mapping should be improved to distinguish the V₂O₅ and rGO.
3. Fig. S10 could be significant. The authors should show the repeated temperature dependency of voltage.
4. The redox potential difference from intrinsic V₂O₅ is interesting. The authors could show the voltage of each electrode from V₂O₅ at varied temperatures.

We thank the editors and reviewers for their time and valuable comments in improving the quality of this manuscript. The manuscript has been revised to reflect the comments of editor and all reviewers. Provided below is our detailed response to each question, and all revised text is marked in blue for convenience.

Reviewer #1

The authors contribute a nice work of zinc ion thermoelectrochemical cells with giant thermopower for low-grade heat harvesting. Based on the thermoextraction effect and thermodiffusion effect, a record-level thermopower of 25.3 mV K^{-1} and an excellent normalized power density of $2.7 \text{ mW m}^{-2} \text{ K}^{-2}$ were achieved. The great performances and the mechanism analyses of $\text{V}_2\text{O}_5@\text{rGO}-1.5$ based ZTECs were demonstrated, and the potential in the application of self-powered electronics was impressive. Moreover, as-constructed ZTECs displayed good energy storage behavior. In my view, such findings in this work are significant and provide a new strategy to fabricate high-performance ionic thermoelectrochemical devices by reasonable electrode engineering. It can be recommended for publication in Nature Communications. However, some specific comments listed below should be further discussed.

Response: We appreciate this referee for his/her time and positive comments on our work.

1. For ionic thermoelectrics, the stability of the measuring the thermopower/Seebeck coefficient is very important. The number of parallel experiments or error bars should be provided.

Response: We thank this referee for this kind suggestion. To well evaluate the Seebeck coefficient of rGO, V_2O_5 , $\text{V}_2\text{O}_5@\text{rGO}-1.0$, $\text{V}_2\text{O}_5@\text{rGO}-1.5$, and $\text{V}_2\text{O}_5@\text{rGO}-2.0$, we have re-fitted and re-tested the thermal charge process systematically. According to the suggestion from this referee, we have profiled the related results with error band or error bar. Benefitting to the adsorption/desorption mechanism of ions, the kinetics of V_2O_5 can be rationally enhanced with the addition of rGO. As summarized in Fig. R1.1a, the $\text{V}_2\text{O}_5@\text{rGO}-1.5$ based ZTEC generated the highest output voltage of $\sim 1.1 \text{ V}$ among all other ZTECs. In fact, such high voltage can be divided as thermal-induced voltage and self-charging induced voltage. To exclude the contribution of self-charge process to the total voltage, we further recorded the voltage curves during self-charging processes in Fig. R1.1b. Owing to the continuous and density framework of rGO, the gradual desorption of ion from interconnected channels endows rGO-contained ZTEC with relatively low self-charging rate. Thus, the $\text{V}_2\text{O}_5@\text{rGO}-1.5$ based ZTEC delivers the highest thermal-induced voltage of 0.72 V . Based on these results, the rGO based ZTEC shows large S_i value of $10.8 \pm 2 \text{ mV K}^{-1}$ based on the ultrafast adsorption/desorption (Fig. R1.1c). While the V_2O_5 based ZTEC only delivers $7.8 \pm 2.6 \text{ mV K}^{-1}$ due to the sluggish insertion/extraction of Zn^{2+} in its crystalline layers. When combining the merits of rGO and V_2O_5 , a giant S_i value of $23.4 \pm 1.5 \text{ mV K}^{-1}$ can be rationally achieved by adjusting the content of rGO. In detail, the slight lower S_i value of $\text{V}_2\text{O}_5@\text{rGO}-1.0$ based ZTEC than that of V_2O_5 based ZTEC may result from the its relatively dense structure and relatively low specific surface area. While for $\text{V}_2\text{O}_5@\text{rGO}-2.0$ based ZTEC, the S_i value is close to that of rGO based ZTEC, which may be caused by the stacking of as-introduced excessive rGO into V_2O_5 . Accordingly, the Seebeck coefficient highly depends on the synergy behavior of thermal adsorption/desorption and thermal insertion/extraction. The superior thermoelectrochemical performance of ZTECs can be realized by the rational heterostructure engineering of rGO and V_2O_5 .

Fig. R1.1. Thermoelectrochemical performances of H-type ZTECs. **a** Thermal charging curves under various temperature differences. **b** Self-charging curves. **c** Thermal-induced voltage under various temperature differences and fitted Seebeck coefficients.

Moreover, we have re-fitted the thermoelectrochemical performances of assembled the quasi-solid-state ZTECs with error band/bar. As shown in Fig. R1.2a, as-assembled device with $V_2O_5@rGO-1.5$ electrode delivers the output voltage of 0.38 V at an ultralow temperature difference of 8 K. After deducting the self-charging contribution, the thermal-induced voltage is 55, 74, 89, 101, 111, 117, 126, and 122 mV at the temperature difference from 1 to 8 K (Fig. R1.2b and c), respectively. Meanwhile, the corresponding Seebeck coefficient of $V_2O_5@rGO-1.5$ based solid-state device is calculated to be around $11.9 \pm 1 \text{ mV K}^{-1}$ (Fig. R1.2c), reflecting good thermal-electrical response in near room temperature low-grade heat harvesting.

Fig. R1.2. Thermoelectrochemical performances of solid-state ZTECs. **a** Thermal charging curves under various temperature differences. **b** Self-charging curves. **c** Thermal-induced voltage under various temperature differences and fitted Seebeck coefficients.

In addition, the abovementioned results and discussion have been revised in the manuscript and Supplementary Information (Page 11-14, 22-23; Page S4, S9, S10, S16, S20; Fig. 2, 5; Supplementary Fig. 15, 17, 27, 32).

2. In the ZTECs, redox reaction occurs on the electrode during thermal charging. How does the electron transfer in the open circuit?

Response: Thanks for this comment. In fact, during the initial stage of thermal charging process, the positive net charges are accumulated on the cold side based on the Soret effect with thermodiffusion of Zn^{2+} and $CF_3SO_3^-$, resulting in the formation of polarized electrical double layers and generating a thermal-induced voltage. Subsequently, the thermo-extraction of Zn^{2+} from cathode like $V_2O_5@rGO-1.5$ and the reduction reaction of Zn^{2+} in Zn anode (plating of Zn^{2+}) are triggered by the positive a of Zn^{2+}/Zn , in which the electrons transport to $V_2O_5@rGO-1.5$ cathode and remove from the Zn anode. The OCV can be further enhanced. In this process, the implementation of transfer electrons in open circuit requires the electrons to move or store on the electrode/electrolyte interface or switch between non-electronic and electronic states in chemical species.

Besides, we have provided this description in the revised manuscript. (Page 5)

3. To clarify the energy conversion mechanism, the morphology change of both cathode and anode under various temperature differences should be provided.

Response: We thank this referee for this kind suggestion. We have observed the morphology change of both $V_2O_5@rGO-1.5$ cathode and Zn-G anode during the thermoelectrochemical measurements by SEM. As shown in Fig. R1.3a1, the $V_2O_5@rGO-1.5$ is evenly wrapped by carbon black and PVDF in the initial electrode. After discharging to 0.2 V, some thin films were covered on the surface of electrode along with the Zn^{2+} insertion to the $V_2O_5@rGO-1.5$ cathode (Fig. R1.3a2), which may be caused by the generation of by-product like $Zn_4(CF_3SO_3)_4(OH)_4 \cdot 3H_2O$. Notably, as-formed film can be gradually decomposed with the increase of temperature difference from 5 to 30 K (Fig. R1.3a3-a8). In line with our previous report (*Adv. Funct. Mater.* 2021, 31, 2006495), the graphite layers are covered on the surface Zn-G (Fig. R1.3b1), implying the successful modification of pure Zn anode. Even discharged to 0.2 V (Zn^{2+} ions are stripped from the Zn-G anode), the graphite layers still can be maintained on the surface of Zn-G (Fig. R1.3b2). Worthily, the morphology of Zn-G anode during the whole thermoelectrochemical process changes obviously. When the temperature difference is 5 K (Fig. R1.3b3), some byproducts nucleate on the surface of Zn-G anode with the plating of Zn^{2+} . Furthermore, the Zn-G anode displays a well-defined array structure formed by numerous thin nanosheets (Fig. R1.3b4-b8). These findings are also confirmed by the results of XRD patterns in Fig. R1.4, which is consistent with previous reports (*Sci. Adv.* 2019, 5, eaax4297; *Chem. Eng. J.* 2022, 442, 136349; *Energy Storage Mater.* 2020, 28, 307-314).

Thus, the possible mechanisms during the whole thermoelectrochemical processes can be concluded as:

Cathode:

Anode:

Fig. R1.3. Morphology changes of electrodes during thermoelectrochemical test. a1-a8 $V_2O_5@rGO-1.5$ cathode and b1-b8 Zn-G anode under various states.

Fig. R1.4. XRD patterns of $V_2O_5@rGO-1.5$ cathode during thermoelectrochemical test.

In addition, we have provided these related contexts and discussion in the revised manuscript and Supplementary Information (Page 19-21; Page S19; Fig. 4; Supplementary Fig. 30; Ref. 51, 52).

4. How about the energy density of constructed ZTECs? Some related results and contents should be added.

Response: Thanks for this comment. We have supplemented the related energy density data of various devices. As shown in Fig. R1.5, the $V_2O_5@rGO-1.5$ based devices exhibits the highest energy density of 265 Wh kg^{-1} at a power density of 151 W kg^{-1} based on the active mass loading of cathode. Even under the highest power density of $12964.3 \text{ W kg}^{-1}$, a high energy density of 84.7 Wh kg^{-1} still can be retained, suggesting the superior energy storage behavior and rate capability of $V_2O_5@rGO-1.5$ based devices.

Fig. R1.5. Ragone plots of as-assembled devices with various cathodes.

Besides, we have provided this related result and description in the revised manuscript and Supplementary Information (Page 17, Page S14, Supplementary Fig. 25).

5. What is the ratio of heat charge part in a fully charged battery with hybrid method?

Response: We thank this referee for this comment. we have carried out the discharge performances of devices after charging with thermal and hybrid process to distinguish the ratio of heat charge part.

Fig. R1.6. The discharge curves of ZTEC with various cathodes recorded after thermal charge and hybrid charge. a rGO . b V_2O_5 . c $V_2O_5@rGO-1.0$. d $V_2O_5@rGO-1.5$. e $V_2O_5@rGO-2.0$.

As profiled in Fig. R1.6, the rGO , V_2O_5 , $V_2O_5@rGO-1.0$, $V_2O_5@rGO-1.5$, and $V_2O_5@rGO-2.0$ based ZTEC exhibits thermo-voltage of approximately 1.2, 1.2, 0.75, 1.1, and 0.7 V with the

temperature difference of 30 K. It is worth mentioning that such output voltage can be further enhanced to 1.6 V by adding power charge. After discharging with a current density of 0.1 A g^{-1} , we can calculate that the ratio of thermal charge part in fully charged rGO, V_2O_5 , $\text{V}_2\text{O}_5@\text{rGO}-1.0$, $\text{V}_2\text{O}_5@\text{rGO}-1.5$, and $\text{V}_2\text{O}_5@\text{rGO}-2.0$ based cell is 55.2%, 68.3%, 41.2%, 69.3%, and 30.0%, respectively. This result indicates that the heat-to-current conversion of ZTEC is highly determined by the kinetics of electrodes.

Moreover, we have added Fig. R1.6 in the manuscript as **Supplementary Fig. 16** and added descriptions about this in the revised manuscript (**Page 12, Page S9**).

6. Besides, can the voltage of ZTECs be maintained continuously if the temperature gradient is eliminated?

Response: We thank this referee for this important comment (*can the heat charged voltage/energy be maintained if the temperature gradient is eliminated*). In order to address the concern of the reviewer on the heat charged voltage/energy, we have supplemented the self-discharging curves of various ZTECs after thermal charging process, as shown in Fig. R1.7. After discharging at a current density of 0.1 A g^{-1} and thermal charge at a temperature difference of 30 K, there is a negligible voltage drop (7~29 mV) occurred when eliminating the temperature gradient. Especially, the $\text{V}_2\text{O}_5@\text{rGO}-x$ based ZTECs shows relatively fast thermoelectrochemical response, and $\text{V}_2\text{O}_5@\text{rGO}-1.5$ based ZTEC delivers the lowest voltage change of $\sim 7 \text{ mV}$, suggesting the dense energy storage behavior of as-proposed zinc ion thermoelectrochemical cell. It is worth mentioning that the gradually voltage growth in self-discharge process with a temperature difference of 0 K can be attributed to the chemically self-charging behaviors (*J. Am. Chem. Soc.* 2021, 143, 37, 15369-15377; *Nat. Commun.* 2020, 11, 2199). This phenomenon is beyond the scope of this work, so we won't discuss it here too much.

Fig. R1.7. The self-discharge curves obtained after thermal charge.

Meanwhile, we have provided above-mentioned contexts in the revised manuscript and Supplementary Information (**Page 12, 13, Page S11, Supplementary Fig. 19**).

Reviewer #2

The manuscript by Li et al. reports a zinc ion thermoelectrochemical cell using the hydrated vanadium pentoxide/rGO ($V_2O_5@rGO$) as the cathode and Zn/graphene as the anode, respectively. The basic concept of the thermoelectrochemical cell has already been reported, and a compositionally very similar cathode (i.e., VO_2 -PC) has also been explored. Though the current work shows enhanced performance, it does not provide new insights nor unexpected results. In addition, the detailed mechanisms are not investigated thoroughly. Many phenomena are not explained. For example, the charging/discharging mechanisms of V_2O_5 cathodes with different concentrations of rGO is are not explored in-depth. The relationship between the Seebeck coefficient of the $V_2O_5@rGO$ and the performance (capacity or lifespan) of the cathodes is unclear. I therefore do not recommend the publication of this manuscript.

Response: Thanks for your time and valuable comments on our work, please see our detailed response below.

1. The novelty of this work should be clearly demonstrated. What are the advantages of V_2O_5 against the VO_2 or other vanadium-based materials should be discussed.

Response: Please see Table R3.3 for a comparison between this work and our previous related study (Z. Li et al. Nat. Commun. 2022, 13, 132.).

Table R3.3. Comparison between this work and our related study.

	Ref: Z. Li et al. Nat. Commun. 2022, 13, 132.	This work	
Electrode structure	 Evenly distributed nanosphere with relatively dense structure by integrating amorphous carbon and electroactive VO_2 nanodots.	 Well-wrapped 1D V_2O_5 nanobelts by 2D rGO layers relatively loose heterostructure.	
Mode	Zinc ion thermoelectrochemical cell		
Interlayer spacing	0.353 nm	~1.1 nm	
Ion diffusion coefficient	$3.16 \times 10^{-11} \text{ cm}^2 \text{ S}^{-1}$	$4.19 \times 10^{-11} \text{ cm}^2 \text{ S}^{-1}$	
Performance	Seebeck coefficient	$\sim 12.5 \text{ mV K}^{-1}$	$23.4 \pm 1.5 \text{ mV K}^{-1}$
	Temperature difference	45 K	30 K
	Voltage difference	0.2-1.01 V	0.2-~1.1 V
	Specific capacity	588 mAh g^{-1} at 0.1 A g^{-1}	375.5 mAh g^{-1} at 0.1 A g^{-1}
	Rate capability	83 mAh g^{-1} at 20 A g^{-1}	145 mAh g^{-1} at 20 A g^{-1}
	Cyclability	15 cycles	60 cycles

In brief, our previous study (Z. Li et al. Nat. Commun. 2022, 13, 132. as Ref. 27 in the manuscript) is about insertion-type cathode (VO_2 -PC) based ZTEC to evaluate the electrochemical performances using various aqueous electrolytes. Although a Seebeck coefficient of 12.5 mV K^{-1} can be obtained

at a temperature difference of 45 K with 0.5 mol L⁻¹ Zn(CF₃SO₃)₂ electrolyte, the performance of heat-to-current in this device is still limited by the large contribution of thermodiffusion of ions in electrolyte and electrode (58.7%). Therefore, the development of high-performance electrode materials with satisfying kinetics behavior for the conversion of low-grade heat near room temperature is a useful approach to improve heat harvesting capability.

In this work, we have proposed a facile heterostructure engineering strategy by hydrated vanadium pentoxide and graphene to enable giant thermopower for zinc ion thermoelectrochemical cells. According to the synergy effect of thermos-extraction/insertion and thermodiffusion process, the kinetics of materials can be one of important role to further enhance the heat-to-current performances. This unique structure of V₂O₅@rGO-1.5 is beneficial to provide continuous pathways, large interlayer spacing, and abundant electroactive sites for Zn²⁺ diffusion and storage. Moreover, an ultrahigh thermopower of 23.4±1.5 mV K⁻¹ and a high output voltage of 1.1 V as well as a superior normalized power density of 1.9 mW m⁻² K⁻² can be impressively obtained by the V₂O₅@rGO-1.5 based ZTECs. Compared to such thermal chargeable supercapacitor in our previous paper referred in the review comment, the performance (such as Seebeck coefficient, output voltage) of ZTECs are greatly improved only by such facile heterostructure engineering.

Besides, we have added the sentences in the introduction of our manuscript to emphasize these differences (**Page 4-5**).

“...As-fabricated ZTEC can covert low-grade heat into electricity with a thermopower of ~12.5 mV K⁻¹ by the combination of thermodiffusion and thermoextraction. Notably, the whole thermoelectrochemical process is dominated by the thermodiffusion effect, which may be caused by the embedding feature of VO₂ in PC matrix. It should be mentioning that the average ion diffusion coefficient of VO₂-PC is in still in a low level (3.16×10⁻¹¹ cm² S⁻¹). To realize a fast thermoelectrochemical response of ZTECs, the kinetics matching of themodiffusion and thermoextraction as well as the ion mobility and conductivity of electrode materials should be well considered. Importantly, vanadium oxides (especially for vanadium pentoxides) are regarded as attractive candidates for energy storage and conversion owing to their relatively large inner spacing, tuned nanostructures, and multielectron reactions....”

2. The structure characterization is very limited. The authors have compared the thermalelectrochemical performance of the V₂O₅@rGO-1.0, V₂O₅@rGO-1.5, and V₂O₅@rGO-2.0, however, the origins account for such disparity are not investigated. Authors only roughly compared the morphology of these cathodes, which provide limited information. Whether these materials possess different electronic structure, coordination environmental, surface area are not clear. It is therefore difficult to understand the relationship between the cathode composition/structure and the performance.

Response: We thank this referee for this important comment. When preparing the raw manuscript, we have put more efforts into the thermoelectrochemical performances of as-obtained electrode materials. So, some information in raw manuscript is limited to well understand the relationship between structure and performance. We totally agree with this referee, the physicochemical properties (*i.e. electronic structure, coordination environmental, surface area*) are highly affect the thermoelectrochemical behaviors of materials. Thus, we have supplemented some key characterizations of V₂O₅, rGO, and their composites to further demonstrate the superiority of our strategy in this work.

Firstly, we have added the XRD patterns of V₂O₅@rGO-1.0 and V₂O₅@rGO-1.5 in the revised

manuscript. As shown in Fig. R2.1a, all the obvious peaks of $V_2O_5@rGO-1.5$ correspond to the characteristic peaks of orthorhombic $V_2O_5 \cdot 1.6H_2O$ (PDF#40-1296). The additional peak around 25.3° in the pattern of $V_2O_5@rGO-1.5$ can be attributed to the (002) plane of amorphous rGO. Interestingly, the pristine V_2O_5 obtained without addition of GO is in terms of orthorhombic V_2O_5 (PDF#89-0612). Notably, the pattern of $V_2O_5@rGO-1.0$ is similar with that of $V_2O_5@rGO-1.5$ (Fig. R2.1b). The interlayer spacing of pure V_2O_5 is significantly enlarged from ~ 0.44 nm to ~ 1.1 nm of $V_2O_5 \cdot 1.6H_2O$ in both $V_2O_5@rGO-1.5$ and $V_2O_5@rGO-1.0$, confirming that acidic GO exhibits important role in regulating the microstructure of vanadium oxides. With the increasing of GO concentration, the $V_2O_5 \cdot 1.6H_2O$ crystals can further evolve into low-valence vanadium species like V_6O_{13} and $V_3O_7 \cdot H_2O$ due to partial reduction of high-valence vanadium species by formed rGO during hydrothermal treatment.

Fig. R2.1. XRD patterns for **a** rGO, V_2O_5 , and $V_2O_5@rGO-1.5$, and **b** $V_2O_5@rGO-1.0$ and $V_2O_5@rGO-2.0$.

Secondly, we have investigated the resistivity as well as the electron conductivity of as-prepared samples. As displayed in Fig. R2.2, the pristine V_2O_5 holds ultralow electron conductivity of $1.4 \times 10^{-5} \text{ S cm}^{-1}$. With the introduction of rGO, the electron conductivity values of $V_2O_5@rGO-x$ composites are significantly enhanced. As expected, the $V_2O_5@rGO-1.5$ exhibits the highest electron conductivity of 1.2 S cm^{-1} , which implies fast transport of electron as well as satisfying power capability in energy conversion and storage.

Fig. R2.2. Resistivity and corresponding electron conductivity of various materials measured by four-point probe method.

Thirdly, we have measured the specific surface area (SSA) of samples by N_2 adsorption-desorption isotherms (Fig. R2.3a). Benefitting to the abundant pores and 3D-interconnected structure, the rGO shows the largest adsorbed quantity among as-prepared materials. Consequently,

a high SSA value of $284.1 \text{ m}^2 \text{ g}^{-1}$ can be achieved (Fig. R2.3b). However, the pristine V_2O_5 only shows $5.2 \text{ m}^2 \text{ g}^{-1}$. After introducing rGO, the SSA value can be reasonably improved to $13.3 \text{ m}^2 \text{ g}^{-1}$. It is worth mentioning that such relatively low SSA values are highly caused by the adopted drying method in an oven ($80 \text{ }^\circ\text{C}$).

Fig. R2.3. **a** N_2 adsorption-desorption isotherms, and **b** corresponding specific surface area values for various materials.

Besides, we have employed the $0.5 \text{ mol L}^{-1} \text{ Zn}(\text{CF}_3\text{SO}_3)_2$ electrolyte to further study the surface wettability of materials by dynamic contact angle meter. In order to avoid the influence of powder voids on the contact angle as much as possible, we prepared various films by using such powder samples. In a typical run, all as-obtained powder samples ($\sim 100 \text{ mg}$) were dispersed into deionized water by ultrasonic treatment, respectively. After filtration and drying, various films formed by each sample can be obtained. As demonstrated in Fig. R2.4a, the contact angle (CA) of pristine V_2O_5 is about 26.1° , suggesting its superhydrophilic feature. Even after standing 20 s, the CA value still can be maintained as 23.2° (Fig. R2.4b). Worthily, the CA value of $\text{V}_2\text{O}_5@\text{rGO}-x$ composites gradually increase with the increase of rGO amount, which may be caused by the stack of graphene layers. According to the change of CA values during immersion, the $\text{V}_2\text{O}_5@\text{rGO}-1.0$ delivers higher electrolyte permeability than that of $\text{V}_2\text{O}_5@\text{rGO}-1.5$ and $\text{V}_2\text{O}_5@\text{rGO}-2.0$. This differences in CA values highly affect to the dissolution of vanadium species from the electrode into the electrolyte, leading to poor durability as well as device failure.

Fig. R2.4. Contact angle analyses for films with various materials. **a** initial state, and **b** after standing 20 s.

Moreover, we have added these related figures and descriptions in the revised manuscript and Supplementary Information to further clarify the relationship between structures and performances

(Page 7-10, Page S3-S7, Supplementary Fig. 4-12).

3. More evidences of the heterostructure should be provided.

Response: Thanks for this comment. Apart from the characterizations presented in Comment#2, we further provided some data including HRTEM, SAED, Raman spectra, XPS spectra to understand the heterostructure.

The crystalline structure of the $V_2O_5@rGO-1.5$ heterostructure was further investigated by HRTEM and selected area electron diffraction (SAED). HRTEM image in Fig. R2.5a clearly indicates the reasonably intertwined microstructure engineered by multilayer rGO and V_2O_5 nanobelts with several micrometers in length, validating the existence of a nanobelt-on-sheet heterostructure in $V_2O_5@rGO-1.5$. Moreover, the ultralarge interlayer spacing of around 1.1 nm can be indexed to the (001) plane of orthorhombic $V_2O_5 \cdot 1.6H_2O$ (Fig. R2.5b), which may greatly boost the ions diffusion kinetics.

Fig. R2.5. Morphology of as-obtained $V_2O_5@rGO-1.5$. **a** HRTEM image. **b** the crystalline structure.

Meanwhile, the SAED pattern of $V_2O_5@rGO-1.5$ exhibits the feature of orthorhombic $V_2O_5 \cdot 1.6H_2O$ and rGO, in consistent with the XRD result (Fig. R2.6). In addition, such SAED pattern also reveals the polycrystalline feature of formed $V_2O_5 \cdot 1.6H_2O$ nanobelts and amorphous rGO layers, further confirming the formation of heterostructure by $V_2O_5 \cdot 1.6H_2O$ and rGO.

Fig. R2.6. The SAED patterns of $V_2O_5@rGO-1.5$.

XPS spectra of V_2O_5 , rGO and $V_2O_5@rGO-x$ composites were further carried out to reveal the integration of carbon matrix and vanadium oxides (Fig. R2.7a). Notably, the broad peak of O 1s for V_2O_5 at 530.3 eV can be attributed to the VO_x (Fig. R2.7b). Deconvolution of O 1s spectra for $V_2O_5@rGO-x$ samples display two additional peaks at 531.4 and 533.1 eV, which corresponds to the C=O groups in rGO and the water in sample, respectively. As shown in Fig. R2.7c, the peaks at 517.3 (V $2p_{3/2}$) and 524.6 eV (V $2p_{1/2}$) for V_2O_5 belong to V^{5+} . After combining with rGO, a part of

V^{4+} can be found in V 2p of $V_2O_5@rGO-x$ composites. Detailly, the ratio of V^{4+} to V^{5+} gradually increases from 0.28 ($V_2O_5@rGO-1.0$) to 0.47 ($V_2O_5@rGO-2.0$), which could be caused by the reduction effect of rGO to V^{5+} during hydrothermal procedure.

Fig. R2.7. XPS analyses for various materials. a XPS survey. High-resolution XPS spectra of b O 1s and c V 2p.

In addition, Fig. R2.8 plots the comparison of Raman spectra for rGO, V_2O_5 , and $V_2O_5@rGO-x$ composites. It is noteworthy that the as-prepared $V_2O_5@rGO-x$ heterostructure exhibits the typical characteristic peaks of both VO_x (142.5 , 194.4 , 662.2 , 404.8 , 698.2 and 992.7 cm^{-1}) and graphene (1330.7 and 1598.2 cm^{-1}), further confirming the composition of the heterostructure.

Fig. R2.8. Raman spectra for various materials.

Overall, the above characterization methods disclose the heterostructure engineered by V_2O_5 nanobelts and graphene layers, which may be conducive to boost ion/charge transport via as-formed

heterointerfaces. At present, we have provided all above-mentioned characterizations and discussion in the revised manuscript and Supplementary Information (**Page 7-10, Page S3-S7, Supplementary Fig. 4-12**).

4. The authors have shown that the output voltages of the various cathodes are significantly different. Do the different contents of rGO affect the thermo adsorption and the ion diffusion? Please give an explanation.

Response: We thank this referee for this key comment. Yes, the content of rGO in $V_2O_5@rGO-x$ composites is one of important roles for ion diffusion as well as the thermoelectrochemical process. As proposed in our work, the ZTECs can convert low-grade heat into electricity based on the synergy effect of thermodiffusion and thermoextraction of ions. Some characteristics like high specific surface area, abundant active sites, and fast diffusion coefficient are highly required for functional electrode materials for harvesting low-grade heat. To further understand the possible effect of introduced rGO, we carried out the specific surface area and Zn^{2+} diffusion coefficient of samples.

As shown in Fig. R2.9, the rGO shows the largest adsorbed quantity among as-prepared materials. Consequently, a high SSA value of $284.1 \text{ m}^2 \text{ g}^{-1}$ can be achieved. However, the pristine V_2O_5 only shows $5.2 \text{ m}^2 \text{ g}^{-1}$. After introducing rGO, the SSA value can be reasonably improved to $13.3 \text{ m}^2 \text{ g}^{-1}$. Notably, the $V_2O_5@rGO-1.5$ holds the largest SSA value among $V_2O_5@rGO-x$ samples, suggesting relatively large space for ions adsorption.

Fig. R2.9. Specific surface area values for various materials.

Meanwhile, we added the galvanostatic intermittent titration techniques (GITT) of V_2O_5 , $V_2O_5@rGO-1.0$, and $V_2O_5@rGO-2.0$ to investigate the Zn^{2+} diffusion coefficient of various electrodes to highlight the structural merits of $V_2O_5@rGO-1.5$ nanocomposite in Zn-based thermoelectrochemical devices. The corresponding GITT curves during charging and discharging processes are displayed in Fig. R2.10. Accordingly, the Zn^{2+} diffusion coefficient (D_{Zn}) of V_2O_5 is in a range of 10^{-12} to $10^{-11} \text{ cm}^2 \text{ s}^{-1}$ during the whole electrochemical tests (Fig. R2.10a and e), demonstrating relatively sluggish insertion/extraction of Zn^{2+} in the pristine V_2O_5 . With the increase of rGO content, the D_{Zn} value can be enhanced to a range of 10^{-11} to $10^{-10} \text{ cm}^2 \text{ s}^{-1}$ for $V_2O_5@rGO-1.0$ and $V_2O_5@rGO-1.5$ (Fig. R2.10b, f and c, g), which is almost an order of magnitude higher than the value of V_2O_5 . However, the D_{Zn} value shows a slight loss in the $V_2O_5@rGO-2.0$ (Fig. R2.10d and h), which mainly caused by the dense wrapping of V_2O_5 nanobelts by stacked rGO layers. As

summarized in Fig. R2.10i, $V_2O_5@rGO-1.5$ exhibits the highest average Zn^{2+} diffusion coefficient during the whole electrochemical processes. Such relatively fast insertion/extraction of Zn^{2+} in the $V_2O_5@rGO-1.5$ electrode may be owing to the relatively large interlayer spacing of $V_2O_5 \cdot 1.6H_2O$ and the high conductivity of rGO substrate.

Fig. R2.10. GITT analyses for various electrodes. **a,e** V_2O_5 . **b,f** $V_2O_5@rGO-1.0$. **c,g** $V_2O_5@rGO-1.5$. **d,h** $V_2O_5@rGO-2.0$. **i** Comparison for the average Zn^{2+} diffusion coefficient.

In addition, we have provided these related results and contexts in the revised manuscript and Supplementary Information (Page 15-16, Page S13, Fig. 3, Supplementary Fig. 23).

5. The authors mentioned that the use of rGO could enhance the stability of V_2O_5 by suppressing the metal dissolution. However, there are not supporting data. They should at least compare the performance of V_2O_5 and rGO and further monitor the metal dissolution rate. Same question goes to the Zn@G anode.

Response: Thanks for the important comment. As suggested by this referee, we have tested the vanadium dissolution rate of V_2O_5 , rGO, $V_2O_5@rGO-x$ electrodes in $0.5 \text{ mol L}^{-1} Zn(CF_3SO_3)_2$ electrolyte by inductively coupled plasma-optical emission spectroscopy (ICP-OES). Fig. R2.11a shows the digital photos of different electrode immersed in 5 mL of electrolyte with different soaking time. Obviously, the electrolyte solution with pristine V_2O_5 shows a noticeable color change from transparent to yellow, suggesting the serious dissolution of V species in electrolyte. As expected, the dissolution of V species of various $V_2O_5@rGO-x$ electrodes can be significantly suppressed by the rGO coating layers. Furthermore, the detail concentration of V species in electrolytes after soaking for 5 days and 10 days is summarized in Fig. R2.11b. After soaking for 5 days, the concentration of V in electrolyte for V_2O_5 is 47.5 mg L^{-1} , which is several times higher than those for $V_2O_5@rGO-x$ electrodes ($8.9\sim 17.3 \text{ mg L}^{-1}$). Even after soaking for 10 days, the average V metal dissolution rate of V_2O_5 (7.1 mg L^{-1} per day) is much higher than those of $V_2O_5@rGO-x$ electrodes ($1.1\sim 2.0 \text{ mg L}^{-1}$ per day), confirming the important role of rGO to maintain the structure stability in $V_2O_5@rGO-x$ composites.

Fig. R2.11. Dissolution of vanadium in 0.5 mol L⁻¹ Zn(CF₃SO₃)₂ electrolyte with different soaking times. a digital photos. **b** ICP-OES results.

In fact, the Zn-G anode used in this work is modified based on our previous reported method (*Adv. Funct. Mater.* 2021, 31, 2006495). After evaluating the plating/stripping behavior of Zn and Zn-G by symmetric cells using 2 mol L⁻¹ ZnSO₄ electrolyte, the functional graphite layer can not only act as ions buffer, but also guide the uniform nucleation of Zn²⁺ in graphite voids. With such synergy effect, the Zn-G displays low overpotential, high reversibility, and dendrite-free durability compared with the pristine Zn. Thus, we employed Zn-G in this work to achieve relatively high durability.

Also, we have immersed Zn and Zn-G electrodes in 0.5 mol L⁻¹ Zn(CF₃SO₃)₂ electrolyte with different soaking time. As shown in Fig. R2.12, the pristine Zn surface was seriously oxidized together with obvious color change. However, the Zn-G still holds relatively stable surface without serious byproducts even after 10 days.

Fig. R2.12. Digital photos of Zn-G and Zn anodes with different soaking times.

Meanwhile, the morphology of Zn foil before/after modification was detected by SEM. Notably, the bare Zn shows a rough surface with numerous uneven defects (Fig. R2.13a1), which is probably formed during the manufacturing procedure. As shown, such uneven surface of pristine Zn was well smoothed by coated graphite layers just relying on physical friction between zinc foil and pencil during drawing (Fig. 2.13b1). After soaking for 10 days, the byproducts pile up and evolve into inter-connected structure in Zn surface, while the surface of Zn-G is still covered by graphite layer with ignorable byproducts (Fig. 2.13a2 and b2). This finding indicates the Zn-G is relatively stable in 0.5 mol L⁻¹ Zn(CF₃SO₃)₂ electrolyte.

Fig. R2.13. SEM images of **a** Zn and **b** Zn-G in initial state (a1, b1) and after soaking 10 days (a2, b2).

Besides, we have newly added this related results in the revised manuscript and Supplementary Information (**Page 10, Page S8, Supplementary Fig. 13**).

6. Will the Seebeck coefficient of $V_2O_5@rGO-1.5$ and the thermo-diffusion/extraction of ions in cathode be affected during cycling at different voltages, given the generation of new phases (e.g., $Zn_xV_2O_5$) and byproducts?

Response: We thank this referee for this comment. In fact, the Seebeck coefficient of is highly determined by electrode and electrolyte. Thus, the generation of new phases and byproducts can affect the Seebeck coefficient as well as the thermos-diffusion/extraction of ions.

To our knowledge, the formation of new phases such as $Zn_xV_2O_5$ and byproducts such as $ZnSO_4[Zn(OH)_2]_3 \cdot xH_2O$ (in $ZnSO_4$ electrolyte) and $Zn(CF_3SO_3)_2[Zn(OH)_2]_3 \cdot xH_2O$ (in $Zn(CF_3SO_3)_2$ electrolyte) is the widely proposed mechanism by the co-insertion of Zn^{2+} and H^+ in V-based zinc ion batteries (*Sci. Adv.*, 2019, 5, eaax4297; *ACS Nano* 2022, 16, 14539-14548; *J. Mater. Chem. A* 2022, 10, 21446-21455). During the thermoelectrochemical tests, the charge carriers like Zn^{2+} and H^+ are preinserted into $V_2O_5@rGO-1.5$ electrode. Similar to the energy storage mechanism of zinc ion batteries, the thermal charge process of zinc ion thermoelectrochemical cell contributed in this work is based on the gradual extraction of ions from $V_2O_5@rGO-1.5$ electrode. However, the thermally charged voltage (~1.1 V) of $V_2O_5@rGO-1.5$ based ZTEC is lower than that of fully charge voltage (1.6 V) of $V_2O_5@rGO-1.5$ based ZIB. Thus, the inserted ions in $V_2O_5@rGO-1.5$ electrode can not be completely extracted, and the byproducts only can be partially decomposed. As pointed in literature (*Chem. Soc. Rev.* 2020, 49, 4203-4219; *ACS Nano* 2022, 16, 14539-14548), the capacity loss of Zn-based devices using vanadium oxides as electrode materials is mainly caused by the vanadium dissolution induced structure destroy and the irreversible byproduct buildup. Under this case, the performances of $V_2O_5@rGO-1.5$ based ZTEC would be greatly limited after several thermal charge and discharge.

As a facile and convenient way to clarify this point, we assembled the cell using various electrodes and adopted long-term cycling tests for about 5000 cycles under a current density of 10

A g^{-1} . After disassembling such cells, the purified electrodes were further used as working electrodes for subsequent thermoelectrochemical measurements. As plotted in Fig. R2.14, the Seebeck coefficient value of each system is relatively lower than that using fresh working electrodes. Especially, the ultralow Seebeck coefficient of $0.75 \pm 0.1 \text{ mV K}^{-1}$ of pristine V_2O_5 is mainly caused by the serious dissolution of vanadium species during long-term cycling tests. Besides, the slightly decrease of Seebeck coefficient value of $\text{V}_2\text{O}_5@\text{rGO}-1.0$, $\text{V}_2\text{O}_5@\text{rGO}-1.5$, $\text{V}_2\text{O}_5@\text{rGO}-2.0$, and rGO could be the influence of byproducts. This result further demonstrates that the rGO coating on V_2O_5 nanobelt shows key role to achieve satisfying durability and stability of vanadium-based devices by suppressing the dissolution of vanadium species to electrolyte.

Fig. R2.14. Seebeck coefficient of ZTECs with different electrodes after long-cycling tests.

Also, we have provided this finding and discussion in the revised manuscript and Supplementary Information (Page 18, Page S16, Supplementary Fig. 27).

7. The Seebeck coefficient of V_2O_5 is significantly enhanced by the rGO. Therefore, the stability and distribution of rGO in the $\text{V}_2\text{O}_5@\text{rGO}$ cathodes, especially after cycling, should be further investigated. Additionally, the Seebeck coefficient of $\text{V}_2\text{O}_5@\text{rGO}-1.5$ (25.3 mV K^{-1}) is close to that rGO (22.4 mV K^{-1}), whereas $\text{V}_2\text{O}_5@\text{rGO}-1.0$ and $\text{V}_2\text{O}_5@\text{rGO}-2.0$ exhibit significantly lower Seebeck coefficient than rGO, why?

Response: Thanks for this comment. According to the kind suggestion from this referee, we have investigated the morphology change and distribution of elements of each cathode after long-term cycling tests by SEM and EDX mapping. The cycled V_2O_5 electrode in $\text{Zn}(\text{CF}_3\text{SO}_3)_2$ electrolyte shows a distinct morphology change due to the obvious vanadium dissolution (Fig. R2.15a). The cycled rGO electrode still holds the inter-connected morphology with abundant pores (Fig. R2.15b), in line with its fresh electrode morphology. In sharp contrast, the cycled $\text{V}_2\text{O}_5@\text{rGO}-x$ electrodes display similar structure and morphology to their pristine electrode (Fig. R2.15c-e), and rGO layers can be visually observed in each electrode.

From the EDX mapping images of each electrode after cycling, we can find that the zinc species

exhibit in all electrodes due to the irreversible electrochemical reactions. Notably, both $V_2O_5@rGO-2.0$ and $V_2O_5@rGO-1.5$ electrodes present the relatively uniform distribution of C, O, Zn, and V element (Fig. R2.15d and e), demonstrating the effect of rGO to suppress the vanadium dissolution. However, the distribution of C and V in cycled $V_2O_5@rGO-1.0$ electrode is relatively uneven, which may be caused by the dissolution of vanadium. Therefore, both the $V_2O_5@rGO-2.0$ and $V_2O_5@rGO-1.5$ electrodes can achieve good stability when comparing with the $V_2O_5@rGO-1.0$ electrode.

Fig. R2.15. SEM images and EDX mapping images of different electrodes after long-cycling tests. a V_2O_5 . b rGO. c $V_2O_5@rGO-1.0$. d $V_2O_5@rGO-1.5$. e $V_2O_5@rGO-2.0$.

To well evaluate the Seebeck coefficient of rGO, V_2O_5 , $V_2O_5@rGO-1.0$, $V_2O_5@rGO-1.5$, and $V_2O_5@rGO-2.0$, we have re-fitted and re-tested the thermal charge process systematically. Such differences of Seebeck coefficient as pointed by this referee may be caused by the fitted error in the original manuscript. According to the suggestion of referees and editors, we have profiled the related results with error band or error bar. Benefitting to the adsorption/desorption mechanism of ions, the kinetics of V_2O_5 can be rationally enhanced with the addition of rGO. As summarized in Fig. R2.16a, the $V_2O_5@rGO-1.5$ based ZTEC generated the highest output voltage of ~ 1.1 V among all other ZTECs. In fact, such high voltage can be divided as thermal-induced voltage and self-charging induced voltage. To exclude the contribution of self-charge process to the total voltage, we further recorded the voltage curves during self-charging processes in Fig. R2.16b. Owing to the continuous and density framework of rGO, the gradual desorption of ion from interconnected channels endows rGO-contained ZTEC with relatively low self-charging rate. Thus, the $V_2O_5@rGO-1.5$ based ZTEC delivers the highest thermal-induced voltage of 0.72 V. Based on these results, the rGO based ZTEC shows large S_i value of 10.8 ± 2 mV K^{-1} based on the ultrafast adsorption/desorption (Fig. R2.16c). While the V_2O_5 based ZTEC only delivers 7.8 ± 2.6 mV K^{-1} due to the sluggish insertion/extraction of Zn^{2+} in its crystalline layers. When combining the merits of rGO and V_2O_5 , a giant S_i value of 23.4 ± 1.5 mV K^{-1} can be rationally achieved by adjusting the content of rGO. In detail, the slight lower S_i value of $V_2O_5@rGO-1.0$ based ZTEC than that of V_2O_5 based ZTEC may result from the its relatively dense structure and relatively low specific surface area. While for $V_2O_5@rGO-2.0$ based ZTEC, the S_i value is close to that of rGO based ZTEC, which may be caused by the stacking of as-introduced excessive rGO into V_2O_5 . Accordingly, the Seebeck coefficient highly depends on the synergy behavior of thermal adsorption/desorption and thermal insertion/extraction. The superior thermoelectrochemical performance of ZTECs can be realized by the rational heterostructure engineering of rGO and V_2O_5 .

Fig. R2.16. Thermoelectrochemical performances of H-type ZTECs. **a** Thermal charging curves under various temperature differences. **b** Self-charging curves. **c** Thermal-induced voltage under various temperature differences and fitted Seebeck coefficients.

In addition, we have revised this description in the manuscript and Supplementary Information (Page 11-13, 17-18; Page S9, S10, S15; Fig. 1; Supplementary Fig. 15, 17, 26).

8. The author should provide more cyclic stability and capacity data of the cells with rGO, V_2O_5 , $V_2O_5@rGO-1.0$, $V_2O_5@rGO-1.5$, and $V_2O_5@rGO-2.0$ cathodes that measured under various temperature differences besides the only one long cycling performance at the room temperature.

Response: Thanks for this important comment. According to this suggestion, we have assembled coin-type cell to further reveal the stability and capability of $V_2O_5@rGO-x$ heterostructure in energy storage under various temperatures (from room temperature to 50 °C). In fact, the capacity and stability of cathodes are highly determined by their physicochemical properties and the vanadium dissolution rate. As shown in Fig. R2.17a, the $V_2O_5@rGO-1.5$ based ZIBs shows superior rate capability at room temperature, in which 375.5, 383.5, 376.9, 353.4, 320.4, 266.3, 208.0, and 145.0 mAh g^{-1} can be recorded at 0.1, 0.2, 0.5, 1.0, 2.0, 5.0, 10.0, and 20.0 A g^{-1} , respectively. Impressively, the reversible discharging capacity can be nearly stabilized at 402.8 mAh g^{-1} as the current density recovers to 0.1 A g^{-1} . The slight increase of specific capacity after test could be caused by the electrolyte penetration induced activation process. The rate performance of $V_2O_5@rGO-1.5$ cathode is also better than that of rGO, V_2O_5 , $V_2O_5@rGO-1.0$, and $V_2O_5@rGO-2.0$ based cells. Fig. R2.17b displays the specific capacity retention of various electrodes at the current density of 10 A g^{-1} . Initially, the capacity of $V_2O_5@rGO-1.5$ increases to 195 mAh g^{-1} due to the gradual activation. Such value can maintain about 122 mAh g^{-1} even over 5000 cycles together with the corresponding Coulombic efficiency of ~100%, suggesting satisfying long cyclic stability. Notably, an obvious increase of capacity before 2000 cycles can be found in V_2O_5 cathode, this may be caused by the electrochemical activation process. However, large capacity decay of V_2O_5 cathode in following cycles may be caused by the dissolution of vanadium. Similar phenomenon can be found in $V_2O_5@rGO-1.0$ cathode. With the increase of adopted temperature (30 °C and 40 °C), the specific capacity of all cathodes can be enhanced and the electrochemical activation process can be accelerated due to the thoroughly infiltration of electrolyte (Fig. R2.17c and d). It is worth mentioning that the $V_2O_5@rGO-2.0$ cathode exhibits better rate capability and cycling stability at 40 °C (Fig. R2.17e and f), which is mainly attributed to its lowest dissolution rate of vanadium into electrolyte among other materials.

Apart from the dissolution of vanadium, the dendrite growth and byproduct formation on Zn-G anode under relatively high temperature (50 °C) become the culprit of cell failure. Worthily, all assembled cells will be short-circuited when the total testing time is around 80 h (Fig. R2.17g and h). Even vanadium dissolution at high temperature will intensify, the cells should be operated with relatively low capacity rather than short-circuited. Thus, the biggest issue of Zn-based cell under high temperature is the serious growth of dendrite.

In this work, we keep the Zn-G anode in cold side during the thermoelectrochemical measurements, only change the temperature of cathode side. This setup also can avoid the serious formation and growth of dendrites in Zn-G surface, which enhances the durability and stability of ZTECs. According to the electrochemical performances obtained from coin-type cells, we can further demonstrate the stability and good rate capability of $V_2O_5@rGO-1.5$ in energy-related applications.

In addition, we have newly added this results and contents in the revised manuscript and Supplementary Information (Page 17-21, Page S14, Supplementary Fig. 24).

Fig. R2.17. Electrochemical performances of ZIBs under different temperatures. a,b Room temperature. **c,d** 30 °C. **e,f** 40 °C. **g,h** 50 °C.

9. The PAM gel is one of excellent materials for sensors, which exhibits high sensitivity for the temperature. Therefore, it is not surprisingly to see the trend shown in Fig. 5b-h. The experimental data of devices with rGO, V_2O_5 , $V_2O_5@rGO-1.0$, and $V_2O_5@rGO-2.0$ cathodes should be also provided for comparison to confirm the enhanced performance of the $V_2O_5@rGO-1.5$ cathode.

Response: We thank this referee for this comment. In fact, the PAM hydrogel is one of widely used gel matrix for quasi-solid-state energy storage devices construction due to its unique mechanical properties and stability. After carefully check, the temperature sensitivity of PAM is rarely reported. To obtain a high sensitivity for the temperature, it is an effective strategy to modify the chain structure of pristine PAM by functional group grafting. For example, the N-isopropylacrylamide (N-IPAM) is widely employed as source to prepare thermosensitive gel (*Adv. Energy Mater.* 2019, 9, 1900433; *Mater. Horiz.* 2021, 8, 1189-1198; *Angew. Chem. Int. Edit.* 2022, 134, e202211132).

Meanwhile, we have carried out the differential scanning calorimetry thermogram (DSC) analysis and in-situ IR spectra to investigate the temperature sensitivity of PAM hydrogel. As shown in Fig. R2.18a, there are no obvious exothermic or endothermic peaks during the adopted temperature range from room temperature to 100 °C, demonstrating the good thermal stability of PAM during thermoelectrochemical tests in this work (temperature difference: 8 K; hot side temperature: 33 °C).

Further, the in-situ IR spectra of as-prepared PAM gel plotted in Fig. R2.18b keep unchanged states, which verifies that the functional groups of PAM gel matrix will not be affected during the thermoelectrochemical measurements.

Fig. R2.18. Characterization of PAM gel. **a** DSC curves from room temperature to 100 °C. **b** In-situ IR spectra under different temperatures from 25 °C to 65 °C.

To highlight the advantages of $V_2O_5@rGO-1.5$, we have assembled the quasi-solid-state ZTECs by using PAM gel electrolyte, Zn-G anode, and as-obtained cathode.

As-assembled device with $V_2O_5@rGO-1.5$ electrode delivers the output voltage of 0.38 V at an ultralow temperature difference of 8 K (Fig. R2.19a). After deducting the self-charging contribution (Fig. R2.19b), the thermal-induced voltage is 55, 74, 89, 101, 111, 117, 126, and 122 mV at the temperature difference from 1 to 8 K, respectively. Meanwhile, the corresponding Seebeck coefficient of $V_2O_5@rGO-1.5$ based solid-state device is calculated to be around $11.9 \pm 1 \text{ mV K}^{-1}$ (Fig. R2.19c), reflecting good thermal-electrical response in near room temperature low-grade heat harvesting among various electrodes.

Fig. R2.19. Thermoelectrochemical performances of quasi-solid-state ZTECs. **a** Thermal charging curves under various temperature differences. **b** Self-charging curves. **c** Thermal-induced voltage under various temperature differences and fitted Seebeck coefficients.

Remarkably, a thermal-induced power density of 0.12 W m^{-2} can be obtained by $V_2O_5@rGO-1.5$ based solid-state ZTEC among various as-constructed ZTECs, signifying an ultrahigh normalized power density of $1.9 \text{ mW m}^{-2} \text{ K}^{-2}$ in state-of-the-art reported systems (Fig. R2.20).

Fig. R2.20. Thermal-induced power density of quasi-solid-state ZTEC.

Inspired by such impressive performances of $V_2O_5@rGO-1.5$ based solid-state ZTEC, an external load with a resistance of $10\text{ k}\Omega$ was employed to examine its stability and durability in energy conversion. As displayed in Fig. R2.21, the output voltage was stably maintained at $\sim 0.4\text{ V}$ with the adoption of temperature gradient, implying the satisfying thermal stability and promise potential of solid-state ZTEC. However, the relatively high voltage decay of rGO and V_2O_5 based solid-state ZTEC are possibly caused by the poor capacity and sluggish ion diffusion kinetics, respectively. In contrast, the slight voltage changes of $V_2O_5@rGO-1.0$ and $V_2O_5@rGO-2.0$ based solid-state ZTEC could be attributed to their relatively low ion diffusion coefficient.

Fig. R2.21. Thermal stability of quasi-solid-state ZTECs with a loading resistance of $10\text{ k}\Omega$.

In addition, we have modified these results according to the kind suggestions from this referee in the revised manuscript and Supplementary Information (Page 22-23; Page S20; Fig. 5; Supplementary Fig. 32).

Reviewer #3

The authors reported exceptionally high thermoelectric power using the mixture of V_2O_5 and rGO. The demonstration seems interesting. However, the manuscript lacks a basic understanding of thermo-electrochemical cells, and the manuscript seems unable to be published.

The fundamental point is that the cell consists of redox-active electrodes, not electrolytes; thus, this is no more TEC. The electric cycle of TEC is achieved by the diffusion of redox-active species (both anolyte and catholyte). Thus, redox-active electrodes are interesting but not suitable for TECs. In other words, the observed electric power could be derived from electrochemical reactions between rGO and V_2O_5 . The authors should show the total cycle of the cell.

Second, such high "thermoelectric voltage" could be derived from overpotentials. The overpotential or mixed potential of electrode materials is sometimes strongly affected by temperature. Thus, cyclic voltammetry or repeated potential measurements should be executed to accurately estimate the effects of these voltages as well as equilibrium potentials. The temperature-dependent CV should be revealed. The repeated thermal charge measurement in Fig. 4a could give important information.

Response: We thank this referee for the important comments. Please see our detailed response below.

In fact, we totally agree with this referee for the view (*The electric cycle of TEC is achieved by the diffusion of redox-active species (both anolyte and catholyte)*). For convenience, we have summarized the characteristics of previous related works (Table R3.1) to give an explanation.

Table R3.1. Comparison of related heat-to-current devices.

TEC system	Thermoelectrics	Thermionic capacitors	Thermocells
Operating mode X. Shi et al. Science 2021, 371, 343.	 Temperature gradient (based on a p-type and an n-type material that drives electrons or holes from source to sink.)	 Temperature gradient (based on separation of positive from negative ions)	 Temperature gradient (based on temperature dependent redox potentials and ions diffusion at the hot and cold sides)
S (mV K^{-1})	~0.1-0.2	~1-10	~1-20
Electrode	Narrow-bandgap semiconductors	MWCNT, metals (Cu, Au, Pt)	MWCNT, metals (Cu, Au, Pt), PANI
Advantage	Continuous operation, high durability	High power density, low cost	High power density, low cost
Limitation	Low efficiency, low temperature difference	Low efficiency, low α , the use of ionic selective membrane	The degradation of long-term cycling

As mentioned in this reference (*X. Shi et al. Science 2021, 371, 343*), thermoelectrics, thermionic capacitors, and thermocells are proposed among the state-of-the-art options for the harvest and direct conversion of low-grade heat into electricity. Obviously, both thermionic capacitors and

thermocells hold the multifold nature during the energy conversion process when compared with the thermoelectricity. In brief, thermionic capacitors involve the ion diffusion driven by temperature difference (ΔT) and the interactions between the accumulated ions and electrodes. For thermocells, such process includes the redox reactions at the electrodes, the ionic diffusion driven by ΔT and chemical reaction-induced ionic concentration gradient, and the interactions between electrolyte and electrodes.

Besides, as-provided work (*H. Im et al. Nat. Commun. 2016, 7, 10600*) is about carbon nanotube aerogel-based thermo-electrochemical cells for the conversion of low-grade waste heat. The planar and cylindrically wound carbon nanotube aerogel sheets are used as thermocell electrodes. 0.4 M aqueous solution of $\text{Fe}(\text{CN})_6^{4-}/\text{Fe}(\text{CN})_6^{3-}$ is employed as electrolyte. According to the reversible redox reactions, as-constructed thermogalvanic cell can convert heat into electricity continuously with temperature gradient. Consequently, a high-power output of 6.6 W m^{-2} with a Carnot-relative efficiency of 3.95% can be obtained. Although this work demonstrates that the electrode purity, engineered porosity and catalytic surfaces play important roles on thermocell performances and parameters. The efficiency and power are still limited by the low surface area of CNT electrodes.

In our work, we have fabricated porous and low-cost electrodes by employing the $\text{V}_2\text{O}_5@\text{rGO}-1.5$ nanocomposite through the integration of electroactive V_2O_5 and conductive porous carbon. According to the thermo-extraction/insertion and thermodiffusion process of insertion-type cathode ($\text{V}_2\text{O}_5@\text{rGO}-1.5$) and stripping/plating behavior of Zn anode, a new system is constructed for low-grade heat conversion and energy storage simultaneously. Compared to related setups based on the mechanisms in Table R3.1, the multivalent feature of charge carriers and insertion process during heat-to-current endow our devices with high capacity and dense charge storage, suggesting its great potential in practical applications.

Meanwhile, we have tested the related cyclic voltammetry (CV) curves of both Zn-G anode and $\text{V}_2\text{O}_5@\text{rGO}-1.5$ cathode using three-electrode system (Fig. R3.1). Notably, the Zn-G anode shows the typical plating/stripping behavior, while the $\text{V}_2\text{O}_5@\text{rGO}-1.5$ cathode presents the reversible redox reaction between vanadium species and zinc ion. According to the CV curves of both electrodes, we can find that the cut-off voltage of Zn-G/ $\text{V}_2\text{O}_5@\text{rGO}-1.5$ could achieve 1.6 V.

Fig. R3.1. CV curves of Zn-G anode and $\text{V}_2\text{O}_5@\text{rGO}-1.5$ cathode recorded by three-electrode system with saturated calomel electrode (SCE) as reference electrode.

The electrochemical behaviors of both anode and cathode during the practical cell are further investigated by the potential change of each electrode, respectively. As shown in Fig. R3.2, the Zn^{2+} can be stripped from Zn-G anode and inserted into $\text{V}_2\text{O}_5@\text{rGO}-1.5$ cathode during the discharge process of Zn-G/ $\text{V}_2\text{O}_5@\text{rGO}-1.5$ full cell. For charge process of Zn-G/ $\text{V}_2\text{O}_5@\text{rGO}-1.5$ full cell,

the inserted ions in $V_2O_5@rGO-1.5$ cathode can be gradually extracted and subsequently plated in Zn-G anode. Based on these processes, the low-grade heat can be converted into electricity and stored in electrodes during the charge part. After discharging by connecting external loads, as-proposed system can be used to further harvest the low-grade heat.

Fig. R3.2. Real-time investigation of voltage for Zn-G anode, $V_2O_5@rGO-1.5$ cathode and full cell.

Indeed, the obtained output voltage combines the thermal voltage with self-charge voltage of electrodes. Thus, we have systematically recorded the thermal charge curve together with the self-charge curves of various systems to achieve a relatively solid result. According to the suggestion of referees and editors, we have profiled the related results with error band or error bar based on the repeated experiment data. Benefitting to the adsorption/desorption mechanism of ions, the kinetics of V_2O_5 can be rationally enhanced with the addition of rGO. As summarized in Fig. R3.3a, the $V_2O_5@rGO-1.5$ based ZTEC generated the highest output voltage of ~ 1.1 V among all other ZTECs. In fact, such high voltage can be divided as thermal-induced voltage and self-charging induced voltage. To exclude the contribution of self-charge process to the total voltage, we further recorded the voltage curves during self-charging processes in Fig. R3.3b. Owing to the continuous and density framework of rGO, the gradual desorption of ion from interconnected channels endows rGO-contained ZTEC with relatively low self-charging rate. Thus, the $V_2O_5@rGO-1.5$ based ZTEC delivers the highest thermal-induced voltage of 0.72 V. Based on these results, the rGO based ZTEC shows large S_i value of 10.8 ± 2 mV K^{-1} based on the ultrafast adsorption/desorption (Fig. R3.3c). While the V_2O_5 based ZTEC only delivers 7.8 ± 2.6 mV K^{-1} due to the sluggish insertion/extraction of Zn^{2+} in its crystalline layers. When combining the merits of rGO and V_2O_5 , a giant S_i value of 23.4 ± 1.5 mV K^{-1} can be rationally achieved by adjusting the content of rGO. In detail, the slight lower S_i value of $V_2O_5@rGO-1.0$ based ZTEC than that of V_2O_5 based ZTEC may result from the its relatively dense structure and relatively low specific surface area. While for $V_2O_5@rGO-2.0$ based ZTEC, the S_i value is close to that of rGO based ZTEC, which may be caused by the stacking of as-introduced excessive rGO into V_2O_5 . Accordingly, the Seebeck coefficient highly depends on the synergy behavior of thermal adsorption/desorption and thermal insertion/extraction. The superior thermoelectrochemical performance of ZTECs can be realized by the rational heterostructure engineering of rGO and V_2O_5 .

Fig. R3.3. Thermochemical performances of H-type ZTECs. a Thermal charging curves under various temperature differences. **b** Self-charging curves. **c** Thermal-induced voltage under various temperature differences and fitted Seebeck coefficients.

According to this kind suggestion (*The temperature-dependent CV should be revealed*), we have tested the related CV curves using H-type cell and coin-type cell, respectively. As summarized in Fig. R3.4, the area enclosed by CV curves of various electrodes enlargers with the increase of adopted temperature difference between anode and cathode, resulting from the completely infiltration of active electrode materials by electrolyte. Notably, the potential values of redox reactions in V_2O_5 and $V_2O_5@rGO-x$ electrodes display slight change. With the increase of the temperature differences, the polarization between oxidized peaks and reduced peaks increases. This phenomenon may be caused by the dendrite growth and byproducts formation on Zn-G anode or the increase of concentration polarization of system.

Fig. R3.4. CV curves of H-type ZTECs under various temperature differences. a rGO. **b** V_2O_5 . **c** $V_2O_5@rGO-1.0$. **d** $V_2O_5@rGO-1.5$. **e** $V_2O_5@rGO-2.0$.

Moreover, we have further measured the temperature-dependent CV curves of each electrode by using coin-type cells. Similar to the results obtained by H-type cells, the area enclosed by CV curves

of various electrodes enlargers with the increase of adopted temperature (Fig. R3.5), which mainly caused by the infiltration of electrolyte ions in cathodes. However, the polarization of redox peaks shows a decrease trend with the increase of temperature. This phenomenon is significantly different with that from H-type cell, and can be attributed to the reduced concentration polarization by rising the temperature of whole cell.

Fig. R3.5. CV curves of coin-type cells under different temperature from RT to 50 °C. a rGO. b V₂O₅. c V₂O₅@rGO-1.0. d V₂O₅@rGO-1.5. e V₂O₅@rGO-2.0.

Overall, the polarization of redox reactions between V₂O₅ and Zn can be significantly optimized by adding moderate rGO. At first, the crystalline structure of orthorhombic V₂O₅ has converted to orthorhombic V₂O₅·1.6H₂O with relatively large interlayer space (~1.1 nm). Secondly, the electron conductivity of pristine V₂O₅ has been enhanced from 1.4×10⁻⁵ to 1.2 S cm⁻¹. Thirdly, the zinc ion diffusion coefficient of V₂O₅ has been improved from 6.7×10⁻¹² to 4.19×10⁻¹¹ cm² S⁻¹. According to the significant enhancement of kinetics, high thermoelectrochemical performances as well as good energy storage behaviors can be realized.

In addition, we have added these related discussions and results in the revised manuscript and Supplementary Information (Page 14, 19; Page S12-13, S17-18; Supplementary Note 1, Supplementary Fig. 21 and 22; Supplementary Note 2, Supplementary Fig. 28 and 29).

The followings are other trivial points to be revised.

1. Figure captions should include more information. Measurement temperature and detailed information on the electrode or solution should be added. The mass of electrodes is important to discuss the relation between the total redox charge and the amount of active materials.

Response: We thank this referee for the careful review. We have thoroughly revised the figure captions in the manuscript and Supplementary Information (Page 8, 14, 21, 23, 25, 26).

2. The resolution of EDX mapping should be improved to distinguish the V₂O₅ and rGO.

Response: Thanks for this comment. We have re-tested the EDX mapping images of V₂O₅@rGO-1.5 materials by employing spherical aberration corrected transmission electron microscope. As shown in Fig. R3.6, the presence of C, V, and O elements in the V₂O₅@rGO-1.5 demonstrates the

formation of heterostructure between V_2O_5 and rGO. Moreover, the V_2O_5 nanobelts are wrapped by the rGO layers. Relatively uniform distribution of C element on V_2O_5 further implies the intimate contact.

Fig. R3.6. Elemental mapping images confirming the presence and distribution of C, V, and O elements in $V_2O_5@rGO-1.5$.

At present, we have modified this related EDX mapping images in the revised manuscript and Supplementary Information (**Page 8, Page S4, Supplementary Fig. 6**).

3. *Fig. S10 could be significant. The authors should show the repeated temperature dependency of voltage.*

Response: We thank this referee for the important comment. At present, we have provided the related data with error band/bar to get a relatively solid result. To well evaluate the Seebeck coefficient of rGO, V_2O_5 , $V_2O_5@rGO-1.0$, $V_2O_5@rGO-1.5$, and $V_2O_5@rGO-2.0$, we have re-fitted and re-tested the thermal charge process systematically.

Benefitting to the adsorption/desorption mechanism of ions, the kinetics of V_2O_5 can be rationally enhanced with the addition of rGO. As summarized in Fig. R3.7a, the $V_2O_5@rGO-1.5$ based ZTEC generated the highest output voltage of ~ 1.1 V among all other ZTECs. In fact, such high voltage can be divided as thermal-induced voltage and self-charging induced voltage. To exclude the contribution of self-charge process to the total voltage, we further recorded the voltage curves during self-charging processes in Fig. R3.7b. Owing to the continuous and density framework of rGO, the gradual desorption of ion from interconnected channels endows rGO-contained ZTEC with relatively low self-charging rate. Thus, the $V_2O_5@rGO-1.5$ based ZTEC delivers the highest thermal-induced voltage of 0.72 V. Based on these results, the rGO based ZTEC shows large S_i value of 10.8 ± 2 $mV K^{-1}$ based on the ultrafast adsorption/desorption (Fig. R3.7c). While the V_2O_5 based ZTEC only delivers 7.8 ± 2.6 $mV K^{-1}$ due to the sluggish insertion/extraction of Zn^{2+} in its crystalline layers. When combining the merits of rGO and V_2O_5 , a giant S_i value of 23.4 ± 1.5 $mV K^{-1}$ can be rationally achieved by adjusting the content of rGO. In detail, the slight lower S_i value of $V_2O_5@rGO-1.0$ based ZTEC than that of V_2O_5 based ZTEC may result from its relatively dense structure and relatively low specific surface area. While for $V_2O_5@rGO-2.0$ based ZTEC, the S_i value is close to that of rGO based ZTEC, which may be caused by the stacking of as-introduced excessive rGO into V_2O_5 . Thus, the Seebeck coefficient highly depends on the synergy behavior of thermal adsorption/desorption and thermal insertion/extraction. The superior thermoelectrochemical performance of ZTECs can be realized by the rational heterostructure engineering of rGO and V_2O_5 .

Fig. R3.7. Thermochemical performances of H-type ZTECs. a Thermal charging curves under various temperature differences. **b** Self-charging curves. **c** Thermal-induced voltage under various temperature differences and fitted Seebeck coefficients.

In addition, we have provided the result discussion in the revised manuscript and Supplementary Information (Page 11-14, 22-23; Page S4, S9, S10, S16, S20; Fig. 2, 5; Supplementary Fig. 15, 17, 27, 32).

4. The redox potential difference from intrinsic V_2O_5 is interesting. The authors could show the voltage of each electrode from V_2O_5 at varied temperatures.

Response: Thanks for this kind suggestion. Accordingly, we have tested the related CV curves using H-type cell and coin-type cell, respectively. As summarized in Fig. R3.8, the area enclosed by CV curves of various electrodes enlargers with the increase of adopted temperature difference between anode and cathode, resulting from the completely infiltration of active electrode materials by electrolyte. Notably, the potential values of redox reactions in V_2O_5 and $V_2O_5@rGO-x$ electrodes display slight change. With the increase of the temperature differences, the polarization between oxidized peaks and reduced peaks increases. This phenomenon may be caused by the dendrite growth and byproducts formation on Zn-G anode or the increase of concentration polarization of system.

Fig. R3.8. CV curves of H-type ZTECs under various temperature differences. a rGO. **b** V_2O_5 . **c** $V_2O_5@rGO-1.0$. **d** $V_2O_5@rGO-1.5$. **e** $V_2O_5@rGO-2.0$.

Moreover, we have further measured the temperature-dependent CV curves of each electrode by using coin-type cells. Similar to the results obtained by H-type cells, the area enclosed by CV curves of various electrodes enlargers with the increase of adopted temperature (Fig. R3.9), which mainly caused by the infiltration of electrolyte ions in cathodes. However, the polarization of redox peaks shows a decrease trend with the increase of temperature. This phenomenon is significantly different with that from H-type cell, and can be attributed to the reduced concentration polarization by rising up the temperature of whole cell.

Fig. R3.9. CV curves of coin-type cells under various temperature differences. **a** rGO. **b** V_2O_5 . **c** $V_2O_5@rGO-1.0$. **d** $V_2O_5@rGO-1.5$. **e** $V_2O_5@rGO-2.0$.

Overall, the polarization of redox reactions between V_2O_5 and Zn can be significantly optimized by adding moderate rGO. At first, the crystalline structure of orthorhombic V_2O_5 has converted to orthorhombic $V_2O_5 \cdot 1.6H_2O$ with relatively large interlayer space (~ 1.1 nm). Secondly, the electron conductivity of pristine V_2O_5 has been enhanced from 1.4×10^{-5} to 1.2 S cm^{-1} . Thirdly, the zinc ion diffusion coefficient of V_2O_5 has been improved from 6.7×10^{-12} to 4.19×10^{-11} cm^2 S^{-1} . According to the significant enhancement of kinetics, high thermoelectrochemical performances as well as good energy storage behaviors can be realized.

Besides, we have added this related data and context in the revised manuscript and Supplementary Information (Page 14, 19; Page S12-13, S17-18; Supplementary Note 1, Supplementary Fig. 21 and 22; Supplementary Note 2, Supplementary Fig. 28 and 29).

Many thanks for the referee's rigorous scholarship and good suggestions. These comments are very valuable, which will improve our research work in future. Many thanks to the editors and these reviewers.

REVIEWER COMMENTS

Reviewer #1 (Remarks to the Author):

The authors have addressed my concerns, and the manuscript is recommended to be published.

Reviewer #2 (Remarks to the Author):

The authors have addressed my comments. The manuscript can be accepted.

Reviewer #3 (Remarks to the Author):

I am impressed with the sincere revisions the authors have made. Unfortunately, however, it is still difficult to make positive comments on this paper.

First, from the temperature-dependent CV, this cell cannot be called a TEC: at a temperature difference of 20°C, this electrode shows a maximum change of only -50 mV. In other words, we can only consider a maximum contribution of -2.5 mV/K.

Secondly, it is incomprehensible to me that a continuous operation can be performed with respect to any other thermoelectric contribution than TEC. This cell seems to be able to operate continuously for about an hour, but in general it is impossible to operate continuously even if the thermal capacitor effect can be temporarily accumulated. It is also the same for the temperature dependence of the reaction of the zinc in the electrodes.

The only remaining possibility seems the temperature dependence of the rate of self-charging. It is not a thermoelectric conversion, but simply the release of chemical energy that the cell possessed from the beginning.

I could not change my mind from this idea according to the response.

We thank the editors and reviewers for their time and valuable comments in improving the quality of this manuscript. The manuscript has been revised to reflect the comments of editor and all reviewers. Provided below is our detailed response to each question, and all revised text is marked in blue for convenience.

Reviewer #1

The authors have addressed my concerns, and the manuscript is recommended to be published.

Response: We appreciate this referee for his/her time and positive comments on our work.

Reviewer #2

The authors have addressed my comments. The manuscript can be accepted.

Response: Thanks for this referee for the encouraging comments on our work.

Reviewer #3

I am impressed with the sincere revisions the authors have made. Unfortunately, however, it is still difficult to make positive comments on this paper.

First, from the temperature-dependent CV, this cell cannot be called a TEC: at a temperature difference of 20°C, this electrode shows a maximum change of only -50 mV. In other words, we can only consider a maximum contribution of -2.5 mV/K.

Secondly, it is incomprehensible to me that a continuous operation can be performed with respect to any other thermoelectric contribution than TEC. This cell seems to be able to operate continuously for about an hour, but in general it is impossible to operate continuously even if the thermal capacitor effect can be temporarily accumulated. It is also the same for the temperature dependence of the reaction of the zinc in the electrodes.

The only remaining possibility seems the temperature dependence of the rate of self-charging. It is not a thermoelectric conversion, but simply the release of chemical energy that the cell possessed from the beginning.

I could not change my mind from this idea according to the response.

Response: We thanks this referee for this important comment. Maybe some of the descriptions in our original manuscript bring you some concerns, sorry for this. Please see the following reply.

Generally, "TEC" represents the typical thermoelectric cell or thermogalvanic cell, which is derived from the traditional solid-state thermoelectric (also called electron-thermoelectric) mainly based on semiconductors. As introduced in the previous literature (*Energy Environ. Sci.*, 2022, 15(9): 3670-3687), the thermo-electrochemical cell (TEC) is always used to infer the chemical systems with ions as energy carriers, such as thermogalvanic cells (TGCs) and thermally chargeable

capacitors (TCCs). Accordingly, we also named our proposed devices, zinc ion thermoelectrochemical cells, as “ZTECs”. Maybe this is not appropriate here. As mentioned in our work and pointed by reviewers, as-proposed devices can realize the energy conversion from low-grade heat to electricity by the synergic effect between thermodiffusion and thermoextraction effects. In fact, the possible chemical reactions are also involved in the electrode/electrolyte interfaces. Thus, we have renamed our devices as zinc ion thermal charging cells (ZTCCs) and revised this thoroughly in the manuscript and Supplementary materials.

We agree with this referee for this point (*at a temperature difference of 20°C, this electrode shows a maximum change of only -50 mV. In other words, we can only consider a maximum contribution of -2.5 mV/K.*). However, the temperature coefficient obtained by the polarization voltage of redox peaks can not represent the whole thermopower/Seebeck coefficient in our work due to the large change of entropy and ions concentration.

For typical TECs, the thermoelectrochemical conversion highly depends on the redox reaction $A_{ox} + n e \leftrightarrow B_{red}$. Under this case, the equilibrium potential (E) of TEC can be described as follows according to the Nernst equation:

$$E = E^0 + \frac{RT}{nF} \ln \frac{(\alpha_{ox})^A}{(\alpha_{red})^B} \quad (R1)$$

where E^0 is the standard potential, R is the ideal gas constant, T is the temperature of electrode, n is the number of electrons transferred in the electrochemical reaction, F is the Faraday constant. The A and B are the coefficients in the redox reaction. α is the activity of the oxidation (ox) or reduction species (red), which can be defined as:

$$\alpha = \gamma C \quad (R2)$$

where γ and C represent the activity coefficient and concentration.

By combing equation R1 and R2, the equation R1 can be rewritten as:

$$E = E^0 + \frac{RT}{nF} \left[\ln \frac{(\gamma_{ox})^A}{(\gamma_{red})^B} + \ln \frac{(C_{ox})^A}{(C_{red})^B} \right] \quad (R3)$$

Moreover, the thermopower of a redox couple can be defined as:

$$S_{TGC} = -\frac{E_H - E_C}{T_H - T_C} \quad (R4)$$

Inspired by the discussion from literatures (*Science*, 2020, 368, 1091-1098; *Sci. Adv.*, 2022, 8, eabl5318), such equation can be further expanded as:

$$S_{TGC} = -\frac{E_H^0 - E_C^0}{T_H - T_C} - \frac{R}{nF\Delta T} \left[T_H \ln \frac{(\gamma_{ox,H})^A}{(\gamma_{red,H})^B} + T_C \ln \frac{(\gamma_{ox,C})^A}{(\gamma_{red,C})^B} \right] - \frac{R}{nF\Delta T} \left[T_H \ln \frac{(C_{ox,H})^A}{(C_{red,H})^B} + T_C \ln \frac{(C_{ox,C})^A}{(C_{red,C})^B} \right] \quad (R5)$$

Here, the temperature coefficient (α_R) can be determined by electrochemistry and defined as equation R6:

$$\alpha_R = \frac{E_H^0 - E_C^0}{T_H - T_C} \quad (R6)$$

Thus, the equation R5 can be modified as follows:

$$S_{TGC} = -\alpha_R - \frac{R}{nF\Delta T} \left[T_H \ln \frac{(\gamma_{ox,H})^A}{(\gamma_{red,H})^B} + T_C \ln \frac{(\gamma_{ox,C})^A}{(\gamma_{red,C})^B} \right] - \frac{R}{nF\Delta T} \left[T_H \ln \frac{(C_{ox,H})^A}{(C_{red,H})^B} + T_C \ln \frac{(C_{ox,C})^A}{(C_{red,C})^B} \right] \quad (R7)$$

As suggested by this referee, it is a good method to distinguish the contribution of redox species on

the thermopower or Seebeck coefficient by the polarization potential of redox peaks in CV curves for typical TECs. It should be pointed out that here mentioned TECs can not exhibit the change of ions concentration. However, due to the exhibition of redox reactions on both electrodes in cold and hot sides, the concentration change of redox species should not be ignored in our ZTCCs. This limitation also clearly shows the difference in typical TECs and our proposed ZTCCs.

As found from the CV results, only a low absolute temperature coefficient of 2.5 mV K^{-1} is obtained for as-developed ZTCC, which is much lower than the Seebeck coefficient ($23.4 \pm 1.5 \text{ mV K}^{-1}$). Such relatively small polarization potential ($\sim 50 \text{ mV}$) indicates the fast electron transfer kinetics and the better reversibility of ZTCCs. Noted from equation R7, in addition to the tested temperature coefficient, some possible factors also could affect thermopower or Seebeck coefficient of ZTCCs. The first one is related to the entropy change of redox species. The second one is the concentration difference of redox species in ZTCCs, which greatly impacts the whole Seebeck coefficient due to the possible chemical reactions between two electrodes. The last one is the thermodiffusion of electrolyte ions in ZTCCs.

Besides, we have provided this part in the revised manuscript and Supplementary materials (**Page 19; Page S17-S19; Supplementary Note 2**).

For the operation mode of TEC and other devices, we totally agree with this referee for this view (*a continuous operation can be performed with respect to any other thermoelectric contribution than TEC*). Till now, the continuous operation and long-term durability are the unique advantages for TECs. It also should mention that the unsatisfying heat-to-current performances of TECs drive researchers to find and develop new materials and devices for the harvesting of low-grade heat. Recently, the thermocells using ions as carriers have drawn much attention. In fact, the limitation for all ionic thermocells including the ZTCCs in our work is the degradation of active species as well as devices during long-term cycling, which also is the biggest issue limited the practical applications of ionic thermoelectric. We have further summarized the performances and durability of other reported systems in literatures. As shown in Table R1, the reported durability of devices is still unsatisfactory.

Table R1. Thermoelectrochemical performances comparison of $\text{V}_2\text{O}_5@\text{rGO}-1.5$ based solid-state ZTCC with other systems.

System	ΔT	$ S $	$P_{\max}/(A \times \Delta T^2)$	Durability	Ref.
Zn-G// $\text{V}_2\text{O}_5@\text{rGO}-1.5$	8	11.9 ± 1	1.9	4000 s	This work
PAAm based TC	10	~ 3	0.66	140 s	Adv. Energy Mater. 12 , 2201542 (2022)
UGdmCl based TGC	10	4.2	1.1	-	Nat. Commun. 9 , 5146 (2018)
3D Au/Cu based i-TE	10	17	0.16	7200 s	Adv. Energy Mater. 12 , 2103666 (2022)
Au/Cu based i-TE	~ 10	12.7	0.66	18000 s	Science 368 , 1091-1098 (2020)
i-PVA based TC	40	-	0.22	3600 s	ACS Nano 16 , 8347-8357 (2022)

Owing to the pristine properties of ionic thermoelectrics, as-proposed devices can harvest the low-grade heat into electricity through quasicontinuous working mode. As mentioned by this referee, we just have recorded the quasicontinuous cycling performance of solid-state ZTCC for 4000 s in the manuscript to highlight the modified durability of devices by electrode engineering. This does not mean that the solid-state ZTCC has failed. It is worth mentioning that the durability of solid-state ZTCC highly depends on the stability of gel electrolyte and active electrodes like zinc anode in this work. In addition, we have retested the quasicontinuous cycling performance of ZTCC after 7 days. As displayed in Fig. R1, the $V_2O_5@rGO-1.5$ based solid-state ZTCC still can realize the thermal charge and discharge processes. This result demonstrates that the ZTCC can be used repetitively for relatively long-term cycling rather than being a one-time energy source. However, the relatively large decay of thermal-induced voltage is mainly caused by the electrode/electrolyte interface deterioration in ZTCCs.

Fig. R1. Thermal stability of solid-state ZTCC after 7 days with a loading resistance of 10 k Ω .

At last, the thermodiffusion of ion carriers as well as the possible thermogalvanic processes under temperature gradient is the basic principle for the ionic thermoelectric. Comparing with the physical procedure (the movement of electrons and holes) in typical semiconductor-based thermoelectrics, the mechanism for ionic thermoelectrics related to the chemical energy release is totally different. As pointed by this referee, the self-charging behavior of ZTCC cannot be completely ignored in this work due to the presence of oxygen in the configuration. In fact, the voltage change of ZTCCs in our work is mainly caused by the extraction of ions from cathode side and plating in the anode side, which is similar to the electrochemically charging process of zinc ion batteries. Besides, we have recorded and deducted the self-charging impact to evaluate the thermoelectrochemical performances of each device in the manuscript. It also should point out that the voltage obtained by self-charging is relatively much lower than that after thermal charging. Therefore, the as-proposed ZTCCs should be one of new-type ionic thermoelectrochemical devices for the harvesting of low-grade heat.

In addition, we have added this related data and context in the revised manuscript and Supplementary Information (**Page 23; Page S21; Supplementary Fig. 33**).

Many thanks for the referee's rigorous scholarship and good suggestions. These comments are very valuable, which will improve our research work in future. Many thanks to the editors and these reviewers.